# Beyond iid weights: sparse and low-rank deep Neural Networks are also Gaussian Processes

**Thiziri Nait Saada, Alireza Naderi & Jared Tanner**
Mathematical Institute
University of Oxford
`{naitsaadat, naderi, tanner}@maths.ox.ac.uk`

## Abstract

The infinitely wide neural network has proven a useful and manageable mathematical model that enables the understanding of many phenomena appearing in deep learning. One example is the convergence of random deep networks to Gaussian Processes that allows a rigorous analysis of the way the choice of activation function and network weights impacts the training dynamics. In this paper, we extend the seminal proof of Matthews et al. (2018) to a larger class of initial weight distributions (which we call Pseudo-iid), including the established cases of iid and orthogonal weights, as well as the emerging low-rank and structured sparse settings celebrated for their computational speed-up benefits. We show that fully connected and convolutional networks initialized with Pseudo-iid distributions are all effectively equivalent up to their variance. Using our results, one can identify the Edge-of-Chaos for a broader class of neural networks and tune them at criticality in order to enhance their training. Moreover, they enable the posterior distribution of Bayesian Neural Networks to be tractable across these various initialization schemes.

## 1 Introduction

Deep neural networks are often studied at random initialization, where in the limit of infinite width, they have been shown to generate intermediate entries which approach Gaussian Processes. Seemingly this was first studied for one-layer networks in Neal (1996) when the weight matrices have identically and independently distributed (iid) Gaussian entries, and became a popular model for deep networks following the seminal results for deep fully connected networks in Lee et al. (2017) and Matthews et al. (2018). Specifically, the latter formulated a proof strategy that quickly became a cornerstone in the field, paving the way for various extensions of the Gaussian Process limit (see Section 1.1). The resulting Gaussian Process has proven to be of practical interest for at least two reasons. First, it emerges as a mathematically motivated choice of prior in the context of Bayesian Neural Networks. Secondly, it helps the analysis of exploding and vanishing gradient phenomena as done in Schoenholz et al. (2017) and Pennington et al. (2018), amongst other network properties.

In this paper, we extend the proof of the Gaussian Process from Matthews et al. (2018) to a larger class of initial weight distributions (which we call Pseudo-iid). Pseudo-iid distributions include the already well-known cases of iid and orthogonal weights. Moreover, we explore other important settings that conform to our more general conditions (e.g. structured sparse and low-rank), yet for which the Gaussian Process limit could not be derived previously due to their violation of the iid assumption.

**Why studying low-rank and structured sparse networks at initialization?** In recent years, deep learning models have significantly increased in size, while concurrently smaller AI chips have been preferred. The most widely studied approaches for bridging the gap between these two aims are to reduce the number of parameters by either pruning the network to be sparse or using low-rank factors. The *lottery ticket hypothesis* Frankle & Carbin (2019) has given empirical evidence of the existence of pruned sub-networks (i.e. *winning tickets*) achieving test accuracy that is comparable and even superior to their original network. Unfortunately most pruning methods produce sparsity patterns

that are unstructured, thereby limiting potential hardware accelerations (see Hoefler et al. (2021)). Remarkably, Chen et al. (2022) revealed some *winning tickets* with structured sparsity, that can be exploited by efficient hardware accelerators on GPUs and FPGAs to reduce greatly their in-memory access, storage, and computational burden (Zhu et al. (2020)). Likewise, the emergence of low-rank structures within deep networks during training is observed where the low-rank factors are aligned with learned classes Han et al. (2022). These factors are shown to improve the efficiency and accuracy of networks, similarly to the lottery ticket hypothesis Osawa et al. (2017); Yang et al. (2020a;b); Tai et al. (2022); Price & Tanner (2023); Karimi Mahabadi et al. (2021); Berlyand et al. (2023). If such computationally efficient *winning tickets* exist, why would one want to train a dense (or full-rank) network at a high computational price only to later reduce it to be sparse (or low-rank)? This question has given rise to single-shot approaches, often referred to as Pruning at Initialization (PaI) such as SNIP Lee et al. (2019), SynFlow Tanaka et al. (2020), GraSP Wang et al. (2020a), NTT Liu & Zenke (2020). Consequently, the present work delves into the properties of such parsimonious networks at initialization, within the larger framework of PSEUDO-IID regimes.

## 1.1 RELATED WORK

To the best of our knowledge, the Gaussian Process behaviour in the infinite width regime was first established by Neal (1996) in the case of one-layer fully connected networks when the weights are IID sampled from standard distributions. The result has then been extended in Matthews et al. (2018) to deep fully connected networks, where the depth is fixed, the weights are distributed as IID Gaussians and the widths of the hidden layers grow jointly to infinity. Jointly scaling the network width substantially distinguishes their method of proof from the approach taken in Lee et al. (2017), where the authors considered a sequential limit analysis through layers. Indeed, analyzing the limiting distribution at one layer when the previous ones have already converged Lee et al. (2017) significantly differs from examining it when the preceding layers are jointly converging to their limits Matthews et al. (2018). The latter approach now serves as a foundational machinery in the infinite width analysis from which numerous subsequent works followed. This paper is one of them.

Since this Gaussian Process behaviour has been established, two main themes of research have further been developed. The first consists of the extension of such results for more general and complex architectures such as convolutional networks with many channels Novak et al. (2020), Garriga-Alonso et al. (2019) or any modern architectures composed of fully connected, convolutional or residual connections, as summarized in Yang (2021), using the Tensor Program terminology. The second line of work concerns the generalization of this Gaussian Process behaviour to other possible weight distributions, such as orthogonal weights in Huang et al. (2021) or, alternatively, any IID weights with finite moments as derived in Hanin (2021). Note that the orthogonal case does not fit into the latter as entries are exchangeable but not independent. Our contribution fits into this line of research, where we relax the independence requirement of the weight matrix entries and instead consider PSEUDO-IID distributions made of uncorrelated and exchangeable random variables. This broader class of distributions, Definition 2, enables us to present a unified proof that strictly generalizes the approaches taken so far and encompasses all of them, for two types of architectures, namely, fully connected and convolutional networks.

## 1.2 ORGANIZATION OF THE PAPER

In Section 2, we focus on fully connected neural networks, formally stating the PSEUDO-IID regime and its associated Gaussian Process (GP) in Theorem 1, whose rigorous proof is provided in Appendix A. Our PSEUDO-IID regime unifies the previously studied settings of IID and orthogonal weights, while also allowing for novel initialization schemes conform to the Gaussian Process limit, such as low-rank and structured sparse weights. We expand the definition of PSEUDO-IID distributions to convolutional kernels and extend the resulting Gaussian Process limit to convolutional neural networks (CNNs) in Section 2.2. In Section 3, we provide examples of PSEUDO-IID distributions in practice, supporting our theoretical results with numerical simulations. Moreover, we allude to two important consequences of the GP limit in 3.3, namely the analysis of stable initialization of deep networks on the so-called Edge-of-Chaos (EoC), as well as tractability for the posterior distribution of Bayesian Neural Networks, in our more expansive PSEUDO-IID regime. Lastly, in Section 4, we review our main contributions and put forward some further research directions.

## 2 Gaussian Process behaviour in the Pseudo-iid regime

We consider an untrained fully connected neural network with width $N_\ell$ at layer $\ell \in \{1, \cdots, L+1\}$. Its weights $W^{(\ell)} \in \mathbb{R}^{N_\ell \times N_{\ell-1}}$ and biases $b^{(\ell)} \in \mathbb{R}^{N_\ell}$ at layer $\ell$ are sampled from centered probability distributions, respectively $\mu_W^{(\ell)}$ and $\mu_b^{(\ell)}$. Starting with such a network, with nonlinear activation $\phi : \mathbb{R} \to \mathbb{R}$, the propagation of any input data vector $z^{(0)} := x \in \mathcal{X} \subseteq \mathbb{R}^{N_0}$ through the network is given by the following equations,

$$h_i^{(\ell)}(x) = \sum_{j=1}^{N_{\ell-1}} W_{ij}^{(\ell)} z_j^{(\ell-1)}(x) + b_i^{(\ell)}, \qquad z_j^{(\ell)}(x) = \phi(h_j^{(\ell)}(x)), \tag{1}$$

where $h^{(\ell)}(x) \in \mathbb{R}^{N_\ell}$ is referred to as the preactivation vector at layer $\ell$, or feature maps.

In this work, we consider all biases to be drawn IID from $\mathcal{N}(0, \sigma_b^2)$ in both fully-connected and convolutional cases. Gaussian biases are in fact needed to ensure the gaussianity of the feature maps. Our focus will thus be restricted to the weight matrices and tensors that result in a Gaussian Process. PSEUDO-IID distributions (formally introduced in Def. 2) only need to meet some moment conditions along with an exchangeability requirement that we define below.

**Definition 1** (Exchangeability). *Let $X_1, \cdots, X_n$ be scalar, vector or matrix-valued random variables. We say $(X_i)_{i=1}^n$ are exchangeable if their joint distribution is invariant under permutations, i.e. $(X_1, \cdots, X_n) \overset{d}{=} (X_{\sigma(1)}, \cdots, X_{\sigma(n)})$ for all permutations $\sigma : [n] \to [n]$. A random matrix is called row- (column-) exchangeable if its rows (columns) are exchangeable random vectors, respectively.*

A row-exchangeable and column-exchangeable weight matrix $W \in \mathbb{R}^{m \times n}$ is not in general entry-wise exchangeable, which means its distribution is not typically invariant under arbitrary permutations of its entries; particularly, out of $(mn)!$ possible permutations of the entries, $W$ only needs to be invariant under $m!n!$ of them — an exponentially smaller number. We can now define the family of PSEUDO-IID distributions.

**Definition 2** (PSEUDO-IID). *Let $m, n$ be two integers and $a \in \mathbb{R}^n$ be any fixed vector. We will say that the random matrix $W = (W_{ij}) \in \mathbb{R}^{m \times n}$ is in the PSEUDO-IID distribution with parameter $\sigma^2$ if*

> *(i) the matrix is row-exchangeable and column-exchangeable,*

> *(ii) its entries are centered, uncorrelated, with variance $\mathbb{E}(W_{ij}^2) = \frac{\sigma^2}{n}$,*

> *(iii) $\mathbb{E}\big| \sum_{j=1}^n a_j W_{ij} \big|^8 = K \|\mathbf{a}\|_2^8 n^{-4}$ for some constant $K$,*

> *(iv) and $\lim_{n \to \infty} \frac{n^2}{\sigma^4} \mathbb{E}(W_{i_a,j} W_{i_b,j} W_{i_c,j'} W_{i_d,j'}) = \delta_{i_a,i_b} \delta_{i_c,i_d}$, for all $j \neq j'$.*

*When $W^{(1)}$ has IID Gaussian entries and the other weight matrices $W^{(\ell)}$, $2 \leq \ell \leq L+1$, of a neural network (see equation 1) are drawn from a PSEUDO-IID distribution, we will say that the network is under the PSEUDO-IID regime.*

The PSEUDO-IID conditions (i)-(iv) are included to serve the following purposes: Conditions (i) and (ii) allow for non-IID initializations such as orthogonal weights while being sufficient for a CLT-type argument to hold. The variance scaling prevents the preactivations in equation 2 from blowing up as the network's size grows. Condition (iii) requires the moments of the possibly dependent weights to behave like independent ones, while condition (iv) constrains their cross-correlations to vanish fast enough. Together they maintain that in the limit, these variables interact as if they were independent, and this is all our proof needs. The importance of condition (iii) is expanded upon in Appendix G. It is worth noting that because the first layer's weight matrix has only one dimension scaling up with $n$, the tension between the entries' dependencies and the network size cannot be resolved, thus we take these weights to be IID [1]. Whether some of the conditions are redundant still remains an open question.

---

[1]In fact, it is enough for $W^{(1)}$ to have IID rows. We assume Gaussian IID entries for simplicity, yet this can be further relaxed at the cost of a more complicated formula for the first covariance kernel (see equation 21).

Throughout this paper, we consider a specific set of activation functions that satisfy the so-called linear envelope property, Definition 3, satisfied by most activation functions used in practice (ReLu, Softmax, Tanh, HTanh, etc.).

**Definition 3.** *(Linear envelope property) A function $\phi : \mathbb{R} \to \mathbb{R}$ is said to satisfy the linear envelope property if there exist $c, M \geq 0$ such that, for any $x \in \mathbb{R}$,*

$$|\phi(x)| \leq c + M|x|.$$

## 2.1 THE PSEUDO-IID REGIME FOR FULLY CONNECTED NETWORKS

Our proofs of PSEUDO-IID networks converging to Gaussian Processes are done in the more sophisticated simultaneous width growth limit as pioneered by Matthews et al. (2018). For a review of the literature on deep networks with sequential vs. simultaneous scaling, see Section 1.1. One way of characterizing such a simultaneous convergence over all layers is to consider that all widths $N_\ell$ are increasing functions of one parameter, let us say $n$, such that, as $n$ grows, all layers' widths increase: $\forall \ell \in \{1, \cdots, L\}, N_\ell := N_\ell[n]$. We emphasize this dependence on $n$ by appending an index $X[n]$ to the random variables $X$ when $n$ is finite and denote by $X[*]$ its limiting random variable, corresponding to $n \to \infty$. The input dimension $N_0$ and the final output layer dimension $N_{L+1}$ are finite thus they do not scale with $n$. Moreover, the input data are assumed to come from a countably infinite input space $\mathcal{X}$ (see A.1). Equation 1 can thus be rewritten, for any $x \in \mathcal{X}$, as

$$h_i^{(\ell)}(x)[n] = \sum_{j=1}^{N_{\ell-1}[n]} W_{ij}^{(\ell)} z_j^{(\ell-1)}(x)[n] + b_i^{(\ell)}, \qquad z_j^{(\ell)}(x)[n] = \phi(h_j^{(\ell)}(x)[n]), \tag{2}$$

and the associated Gaussian Process limit is given in Theorem 1, which we can now state.

**Theorem 1** (GP limit for fully connected PSEUDO-IID networks). *Suppose a fully connected neural network as in equation 2 is under the PSEUDO-IID regime with parameter $\sigma_W^2$ and the activation satisfies the linear envelope property Def. 3. Let $\mathcal{X}$ be a countably-infinite set of inputs. Then, for every layer $2 \leq \ell \leq L + 1$, the sequence of random fields $(i, x) \in [N_\ell] \times \mathcal{X} \mapsto h_i^{(\ell)}(x)[n] \in \mathbb{R}^{N_\ell}$ converges in distribution to a centered Gaussian Process $(i, x) \in [N_\ell] \times \mathcal{X} \mapsto h_i^{(\ell)}(x)[*] \in \mathbb{R}^{N_\ell}$, whose covariance function is given by*

$$\mathbb{E}\left[h_i^{(\ell)}(x)[*] \cdot h_j^{(\ell)}(x')[*]\right] = \delta_{i,j} K^{(\ell)}(x, x'), \tag{3}$$

*where*

$$K^{(\ell)}(x, x') = \begin{cases} \sigma_b^2 + \sigma_W^2 \mathbb{E}_{(u,v) \sim \mathcal{N}(\mathbf{0}, K^{(\ell-1)}(x,x'))}[\phi(u)\phi(v)], & \ell \geq 2, \\ \sigma_b^2 + \frac{\sigma_W^2}{N_0} \langle x, x' \rangle, & \ell = 1. \end{cases} \tag{4}$$

## 2.2 THE PSEUDO-IID REGIME FOR CONVOLUTION NEURAL NETWORKS

We consider a CNN with $C_\ell$ number of channels at layer $\ell \in \{1, \cdots, L + 1\}$ and two-dimensional convolutional filters $\mathbf{U}_{i,j}^{(\ell)} \in \mathbb{R}^{k \times k}$ mapping the input channel $j \in \{1, \cdots, C_{\ell-1}\}$ to the output channel $i \in \{1, \cdots, C_\ell\}$. The input signal $\mathbf{X}$ (also two-dimensional) has $C_0$ channels $\mathbf{x}_1, \cdots, \mathbf{x}_{C_0}$ and its propagation through the network is given by

$$\mathbf{h}_i^{(\ell)}(\mathbf{X})[n] = \begin{cases} b_i^{(1)}\mathbf{1} + \sum_{j=1}^{C_0} \mathbf{U}_{i,j}^{(1)} \star \mathbf{x}_j, & \ell = 1, \\ b_i^{(\ell)}\mathbf{1} + \sum_{j=1}^{C_{\ell-1}} \mathbf{U}_{i,j}^{(\ell)} \star \mathbf{z}_j^{(\ell-1)}(\mathbf{X})[n], & \ell \geq 2, \end{cases} \qquad \mathbf{z}_i^{(\ell)}(\mathbf{X})[n] = \phi(\mathbf{h}_i^{(\ell)}(\mathbf{X}))[n]. \tag{5}$$

In equation 5, the symbol $\mathbf{1}$ should be understood as having the same size as the convolution output, the non-linearity $\phi(\cdot)$ is applied entry-wise, and we emphasize on the simultaneous scaling with $n$ by appending $[n]$ to the feature maps $\mathbf{h}_i^{(\ell)}(\mathbf{X})[n]$. We denote spatial (multi-) indices by boldface Greek letters $\boldsymbol{\mu}, \boldsymbol{\nu}$, etc., that are ordered pairs of integers taking values in the range of the size of the array.

For example, if $\mathbf{X}$ is an RGB ($C_0 = 3$) image of $H \times D$ pixels, $i = 2$, and $\boldsymbol{\mu} = (\alpha, \beta)$, then $X_{i,\boldsymbol{\mu}}$ returns the green intensity of the $(\alpha, \beta)$ pixel. Moreover, we define $[\![\boldsymbol{\mu}]\!]$ to be the patch centered at the pixel $\boldsymbol{\mu}$ covered by the filter, e.g. if $\boldsymbol{\mu} = (\alpha, \beta)$ and the filter covers $k \times k = (2k_0 + 1) \times (2k_0 + 1)$ pixels, then $[\![\boldsymbol{\mu}]\!] = \{(\alpha', \beta') \mid \alpha - k_0 \leq \alpha' \leq \alpha + k_0, \ \beta - k_0 \leq \beta' \leq \beta + k_0\}$, with the usual convention of zero-padding for the out-of-range indices. Sufficient conditions for PSEUDO-IID CNNs to converge to a Gaussian Process in the simultaneous scaling limit are given in Definition 4.

**Definition 4** (PSEUDO-IID for CNNs). *Consider a CNN with random filters and biases $\{\mathbf{U}_{i,j}^{(\ell)}\}$ and $\{b_i^{(\ell)}\}$ as in equation 5. It is said to be in the PSEUDO-IID regime with parameter $\sigma^2$ if $\mathbf{U}^{(1)}$ has IID $\mathcal{N}(0, \frac{\sigma^2}{C_0})$ entries and, for $2 \leq \ell \leq L + 1$, such that for any fixed vector $\mathbf{a} := (a_{j,\boldsymbol{\nu}})_{j,\boldsymbol{\nu}}$,*

    *(i) the convolutional kernel $\mathbf{U}^{(\ell)} \in \mathbb{R}^{C_\ell \times C_{\ell-1} \times k \times k}$ is row-exchangeable and column-exchangeable, that is its distribution is invariant under permutations of the first and second indices,*

    *(ii) the filters' entries are centered, uncorrelated, with variance $\mathbb{E}[(\mathbf{U}_{i,j,\boldsymbol{\mu}}^{(\ell)})^2] = \sigma^2/C_{\ell-1}$,*

    *(iii) $\mathbb{E}\big| \sum_{j=1}^{C_{\ell-1}} \sum_{\boldsymbol{\nu}} a_{j,\boldsymbol{\nu}} \mathbf{U}_{i,j,\boldsymbol{\nu}}^{(\ell)} \big|^8 = K \|\mathbf{a}\|_2^8 (C_{\ell-1})^{-4}$ for some constant $K$,*

    *(iv) and $\lim_{n \to \infty} \frac{C_{\ell-1}[n]^2}{\sigma^4} \mathbb{E}\big(\mathbf{U}_{i_a,j,\boldsymbol{\mu}_a}^{(\ell)} \mathbf{U}_{i_b,j,\boldsymbol{\mu}_b}^{(\ell)} \mathbf{U}_{i_c,j',\boldsymbol{\mu}_c}^{(\ell)} \mathbf{U}_{i_d,j',\boldsymbol{\mu}_d}^{(\ell)}\big) = \delta_{i_a,i_b} \delta_{i_c,i_d} \delta_{\boldsymbol{\mu}_a,\boldsymbol{\mu}_b} \delta_{\boldsymbol{\mu}_c,\boldsymbol{\mu}_d}$, for all $j \neq j'$.*

Similar to the fully connected case, the filters' entries can now exhibit some interdependencies as long as their cross-correlations vanish at a fast enough rate dictated by conditions (iii)-(iv).

**Theorem 2** (GP limit for CNN PSEUDO-IID networks). *Suppose a CNN as in equation 5 is under the PSEUDO-IID regime with parameter $\sigma_W^2$ and the activation satisfies the linear envelope property Def. 3. Let $\mathcal{X}$ be a countably-infinite set of inputs and $\boldsymbol{\mu} \in \mathcal{I}$ denote a spatial (multi-) index. Then, for every layer $1 \leq \ell \leq L + 1$, the sequence of random fields $(i, \mathbf{X}, \boldsymbol{\mu}) \in [C_\ell] \times \mathcal{X} \times \mathcal{I} \mapsto h_{i,\boldsymbol{\mu}}^{(\ell)}(\mathbf{X})[n]$ converges in distribution to a centered Gaussian Process $(i, \mathbf{X}, \boldsymbol{\mu}) \in [C_\ell] \times \mathcal{X} \times \mathcal{I} \mapsto h_{i,\boldsymbol{\mu}}^{(\ell)}(\mathbf{X})[*]$, whose covariance function is given by*

$$\mathbb{E}\left[ h_{i,\boldsymbol{\mu}}^{(\ell)}(\mathbf{X})[*] \cdot h_{j,\boldsymbol{\mu}'}^{(\ell)}(\mathbf{X}')[*] \right] = \delta_{i,j} \Big( \sigma_b^2 + \sigma_W^2 \sum_{\boldsymbol{\nu} \in [\![\boldsymbol{\mu}]\!] \cap [\![\boldsymbol{\mu}']\!]} K_{\boldsymbol{\nu}}^{(\ell)}(\mathbf{X}, \mathbf{X}') \Big), \tag{6}$$

*where*

$$K_{\boldsymbol{\nu}}^{(\ell)}(\mathbf{X}, \mathbf{X}') = \begin{cases} \mathbb{E}_{(u,v) \sim \mathcal{N}(\mathbf{0}, K_{\boldsymbol{\nu}}^{(\ell-1)}(\mathbf{X},\mathbf{X}'))}[\phi(u)\phi(v)], & \ell \geq 2, \\ \frac{1}{C_0} \sum_{i=1}^{C_0} X_{i,\boldsymbol{\nu}} X_{i,\boldsymbol{\nu}}', & \ell = 1. \end{cases} \tag{7}$$

These equations resemble those derived in the fully connected case, apart from the introduction of additional terms accounting for pixels within patch $\boldsymbol{\mu}$. These terms vanish when the filter kernel size is reduced to $k = 1$. Our proof of Theorem 2 can be found in Appendix B.

## 3 PSEUDO-IID IN PRACTICE

In this section, we illustrate some important PSEUDO-IID distributions through examples. We further show that the Gaussian behaviour can serve, among others, two purposes of practical significance (see section 3.3). First, we elucidate the analysis of signal propagation in neural networks. Secondly, this Gaussian Process limit simplifies the initially complicated task of choosing a prior over a large set of distributions a Bayesian Neural Network could realize at initialization.

### 3.1 EXAMPLES OF PSEUDO-IID DISTRIBUTIONS

We elaborate here on some typical initialization schemes and their belonging to the PSEUDO-IID class. To the best of our knowledge, rigorous proofs of the Gaussian Process convergence in the literature are restricted to the IID cases only (see Section 1.1). An important non-IID case that also results in a Gaussian Process is the random orthogonal initialization, partially derived in Huang et al. (2021)

for fully connected networks. Nonetheless, we regret the authors remained elusive on the treatment of the first layer, an aspect which we have since then enhanced. Our proposed PSEUDO-IID regime encompasses both IID and orthogonal cases (see Appendix D) but also allows for a broader class of weight distributions such as random low-rank or structured sparse matrices, for which the Gaussian Process limit has not been established before, despite their practical significance (see Section 1).

**Low-rank weights.** Low-rank structures are widely recognized for speeding up matrix multiplications and can be used to reduce memory requirements, see 1. Whilst such structures inevitably impose dependencies between the weight matrix entries $A \in \mathbb{R}^{m \times n}$, thus breaking the IID assumption, Nait Saada & Tanner (2023) introduced a low-rank framework that falls within our PSEUDO-IID regime. Let $C := [C_1, \cdots, C_r] \in \mathbb{R}^{m \times r}$ be a uniformly drawn orthonormal basis for a random $r$-dimensional subspace. Suppose $P = (P_{ij}) \in \mathbb{R}^{r \times n}$ has IID entries $P_{ij} \overset{\text{iid}}{\sim} \mathcal{D}$. Setting $A := CP$, the columns of $A$ are spanned by those of $C$. The row and column exchangeability of $A$ follows immediately from that of $C$ and $P$, and the moment conditions (iii) and (iv) are controlled by the choice of distribution $\mathcal{D}$. Direct computation of the four-cross product that appears in condition (iv) gives us

$$\mathbb{E}(A_{i_a,1} A_{i_b,1} A_{i_c,2} A_{i_d,2}) = s^2 \sum_{1 \leq k,k' \leq r} \mathbb{E}\Big[ C_{i_a,k} C_{i_b,k} C_{i_c,k'} C_{i_d,k'} \Big],$$

where $s := \mathbb{E}(P_{1,1}^2)$. Using the expression in Lemma 3 of Huang et al. (2021) we can calculate the above expectation and deduce condition (iv) when $r$ is linearly proportional to $m$.

**Structured sparse weights.** Block-wise pruned networks have recently been under extensive study for their efficient hardware implementation Dao et al. (2022b;a). Once the sparsifying mask is fixed, we may apply random row and column permutations on the weight matrices without compromising the accuracy or the computational benefit. Let $A = (A_{ij}) \in \mathbb{R}^{m \times n}$ have IID entries $A_{ij} \overset{\text{iid}}{\sim} \mathcal{D}$ and $B \in \mathbb{R}^{m \times n}$ be the binary block-sparse mask. Let $\tilde{A} = P_m(A \odot B)P_n$, where $P_m$ and $P_n$ are random permutation matrices of size $m \times m$ and $n \times n$ respectively, and $\odot$ represents entry-wise multiplication. Then, by construction, $\tilde{A}$ is row- and column-exchangeable and, for suitable choices of underlying distribution $\mathcal{D}$, it satisfies the moment conditions of Definition 2. Appendix H contains some illustrations for reference.

**Orthogonal CNN filters.** Unlike the fully connected case, it is not obvious how to define the orthogonality of a convolutional layer and, once defined, how to randomly generate such layers for initialization. Xiao et al. (2018) defines an orthogonal convolutional kernel $\mathbf{U} \in \mathbb{R}^{c_{out} \times c_{in} \times k \times k}$ made of $c_{out}$ filters of size $k \times k$ through the energy preserving property $\|\mathbf{U} \star \mathbf{X}\|_2 = \|\mathbf{X}\|_2$, for any signal $\mathbf{X}$ with $c_{in}$ input channels. Wang et al. (2020b) requires the matricized version of the kernel to be orthogonal, while Qi et al. (2020) gives a more stringent definition imposing isometry, i.e.

$$\sum_{i=1}^{c_{out}} \mathbf{U}_{i,j} \star \mathbf{U}_{i,j'} = \begin{cases} \boldsymbol{\delta}, & j = j', \\ 0, & \text{otherwise,} \end{cases}$$

where $\boldsymbol{\delta}$ is 1 at $(0,0)$, and 0 elsewhere. Another definition in Huang et al. (2021) calls for orthogonality of "spatial" slices $\mathbf{U}_{\boldsymbol{\mu}} \in \mathbb{R}^{c_{out} \times c_{in}}$, for all positions $\boldsymbol{\mu}$.

We take a different approach than Wang et al. (2020b) for matricizing the tensor convolution operator, setting the stride to 1 and padding to 0: reshape the kernel $\mathbf{U}$ into a matrix $\tilde{\mathbf{U}} \in \mathbb{R}^{c_{out} \times k^2 c_{in}}$ and unfold the signal $\mathbf{X}$ into $\tilde{\mathbf{X}} \in \mathbb{R}^{k^2 c_{in} \times d}$, where $d$ is the number of patches depending on the sizes of the signal and the filter. This allows $\tilde{\mathbf{U}}$ to be an arbitrary unstructured matrix rather than the doubly block-Toeplitz matrix considered in Wang et al. (2020b), as shown in Figure 1. Matricizing the tensor convolution operator imposes the structure on the signal $\tilde{\mathbf{X}}$ rather than the filter $\tilde{\mathbf{U}}$. Orthogonal (i.e. energy-preserving) kernels $\tilde{\mathbf{U}}$ can then be drawn uniformly random with orthogonal columns, such that

$$\tilde{\mathbf{U}}^\top \tilde{\mathbf{U}} = \frac{1}{k^2} I \tag{8}$$

and then reshaped into the original tensor kernel $\mathbf{U}$. Note that this construction is only possible when $\tilde{\mathbf{U}}$ is a tall matrix with trivial null space, that is when $c_{out} \geq k^2 c_{in}$, otherwise the transpose might be

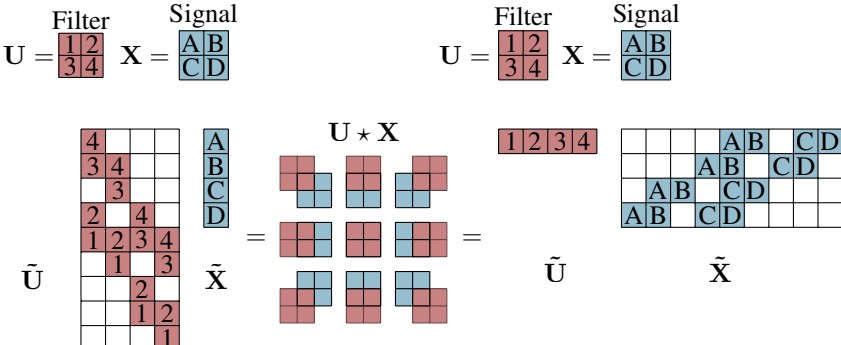

Figure 1: There are various ways to compute convolutions $\mathbf{U} \star \mathbf{X}$ between a tensor filter $\mathbf{U}$ and a 2D signal $\mathbf{X}$ (in the middle) from matrix multiplications. We illustrate the approach taken in Garriga-Alonso et al. (2019) on the left, where the reshaping procedure is applied to the filter, whilst the method we followed, shown on the right, consists of reshaping the signal instead in order to define special structures on the CNN filters such as orthogonality, sparsity and low-rank.

considered. We emphasize that equation 8 is a sufficient (and not necessary) condition for $\mathbf{U}$ to be energy-preserving, since $\tilde{\mathbf{X}}$, by construction, belongs to a very specific structure set $T \subseteq \mathbb{R}^{k^2 c_{in} \times d}$, and, therefore, $\tilde{\mathbf{U}}$ needs to preserve norm only on $T$ (so not everywhere). Therefore, we do not claim the generated orthogonal convolutional kernel $\mathbf{U}$ is "uniformly distributed" over the set of all such kernels.

Now let us check that what we defined to be orthogonal filters in the previous section verify the conditions of Definition 4. Each filter $\mathbf{U}_{i,j}$ is flattened as $\tilde{\mathbf{U}}_{i,j} \in \mathbb{R}^{1 \times k^2}$ and forms part of a row of $\tilde{\mathbf{U}}$ as shown below:

$$
\tilde{\mathbf{U}} = \begin{bmatrix} \tilde{\mathbf{U}}_{1,1} & \tilde{\mathbf{U}}_{1,2} & \cdots & \tilde{\mathbf{U}}_{1,c_{in}} \\ \tilde{\mathbf{U}}_{2,1} & \tilde{\mathbf{U}}_{2,2} & \cdots & \tilde{\mathbf{U}}_{2,c_{in}} \\ \vdots & \vdots & \ddots & \vdots \\ \tilde{\mathbf{U}}_{c_{out},1} & \tilde{\mathbf{U}}_{c_{out},2} & \cdots & \tilde{\mathbf{U}}_{c_{out},c_{in}} \end{bmatrix}. \tag{9}
$$

Applying permutations on the indices $i$ and $j$ translates to permuting rows and "column blocks" of the orthogonal matrix $\tilde{\mathbf{U}}$, which does not affect the joint distribution of its entries. Hence, the kernel's distribution is unaffected, and therefore $\mathbf{U}$ is row- and column-exchangeable. The moment conditions are both straightforward to check as $\mathbf{U}_{i,j,\boldsymbol{\mu}} = \tilde{\mathbf{U}}_{i,(j-1)k^2+\mu}$, $\mu \in \{1, \cdots, k^2\}$, where $\mu$ is the counting number of the pixel $\boldsymbol{\mu}$. To check condition (iv), note that $\mathbb{E}\big(\mathbf{U}^{(\ell)}_{i_a,1,\boldsymbol{\mu}_a} \mathbf{U}^{(\ell)}_{i_b,1,\boldsymbol{\mu}_b} \mathbf{U}^{(\ell)}_{i_c,2,\boldsymbol{\mu}_c} \mathbf{U}^{(\ell)}_{i_d,2,\boldsymbol{\mu}_d}\big) = \mathbb{E}\big(\tilde{\mathbf{U}}_{i_a,\mu_a} \tilde{\mathbf{U}}_{i_b,\mu_b} \tilde{\mathbf{U}}_{i_c,k^2+\mu_c} \tilde{\mathbf{U}}_{i_d,k^2+\mu_d}\big)$, that is a four-cross product of the entries of an orthogonal matrix, whose expectation is explicitly known to be $\frac{C_\ell+1}{(C_\ell-1)C_\ell(C_\ell+2)} \delta_{i_a,i_b} \delta_{i_c,i_d} \delta_{\mu_a,\mu_b} \delta_{\mu_c,\mu_d}$ (Huang et al., 2021, Lemma 3).

### 3.2 SIMULATIONS OF THE GAUSSIAN PROCESSES IN THEOREM 1 FOR FULLY CONNECTED NETWORKS WITH PSEUDO-IID WEIGHTS

Theorem 1 establishes that random fully connected PSEUDO-IID networks converge to Gaussian Processes in the infinite width limit. Here we conduct numerical simulations which validate this for modest dimensions of width $N_\ell = n$ for $n = 3, 30$, and $300$. Figure 2 shows histograms of the empirical distribution of the preactivations when the weights are drawn from various PSEUDO-IID distributions.[2] Even at $n = 30$ there is excellent agreement of the histograms with the limiting

---

[2]Experiments conducted for Figures 2-5 used a fully connected network with activation $\phi(x) = \tanh(x)$, weight variance $\sigma_w = 2$ and without bias. Dropout used probability $1/2$ of setting an entry to zero, low-rank used rank $\lceil n/2 \rceil$, and block-sparsity used randomly permuted block-diagonal matrices with block-size $\lceil n/5 \rceil$. The code to reproduce all these figures can be found at https://shorturl.at/gNOQ0.

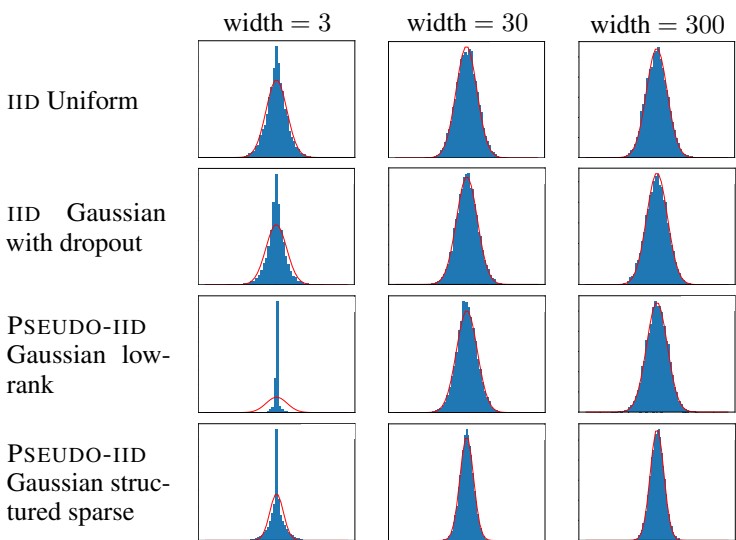

Figure 2: For different instances of the PSEUDO-IID regime, in the limit, the preactivation given in the first neuron at the fifth layer tends to a Gausssian whose moments are given by Theorem 1. The experiments were conducted 10000 times on a random 7-layer deep fully connected network with input data sampled from $\mathbb{S}^8$.

variance predicted by our Theorem 1. These histograms in Figure 2 are further quantified with Q-Q plots in Appendix E.

Figure 3 illustrates the rate with which two independent inputs $x_a$ and $x_b$ generate feature maps corresponding to uncorrelated Gaussian Processes, when passing through a network initialized with PSEUDO-IID weights. The experiment is done by examining the joint empirical distribution of $h_i^{(\ell)}(x_a)[n]$ and $h_i^{(\ell)}(x_b)[n]$ at the same neuron index $i$ and layer $\ell$ through the same network. The level curves depict the theoretical joint Gaussian distribution given by the covariance kernel in Theorem 1. Interestingly, we observe the fastest convergence to the Gaussian Process in the orthogonal initialization. The other PSEUDO-IID distributions considered exhibit good agreement at $n = 30$ which improves at $n = 300$. Some more experiments for fully connected networks and CNNs initialized with orthogonal filters can be found in respectively, Appendix E and F.

### 3.3    IMPLICATIONS OF THE GAUSSIAN PROCESS LIMIT

**Bayesian Neural Network and Gaussian Process.**    As opposed to frequentist approaches, Bayesian inference aims at finding a predictive distribution that serves to quantify the model's predictions uncertainty and prevent overfitting. Starting from a prior, this posterior distribution is updated once the data is observed. Considering Bayesian neural networks, these priors are to be defined on the weights and biases if adopting a parametric perspective, or, alternatively, directly on the input-output function represented by the model. Note how one can jump from one view to the other given that random initialization of the parameters of a neural network induces a distribution on the input-output function $h^{(L+1)}(x)$ of the network. Nonetheless, finding a meaningful prior over the space of functions is in general a delicate task.

In the large width limit, when the weights are drawn PSEUDO-IID in a $L$-layer deep network, Theorem 1 secures that the function represented by the network is *exactly* a Gaussian Process with known covariance kernel $K^{(L+1)}$, which only depends on the variances $\sigma_b, \sigma_W$ and the activation function. Therefore, the exact posterior distribution can be computed, yielding the Neural Network Gaussian Process (NNGP) equivalence described in Novak et al. (2020). Since the covariance kernel obtained in our less restrictive setting recovers the one in Novak et al. (2020), Matthews et al. (2018) and Garriga-Alonso et al. (2019), their experiments still hold in our case and we refer to these works for practical considerations on how to compute the posterior distribution. Specifically, NNGP yields

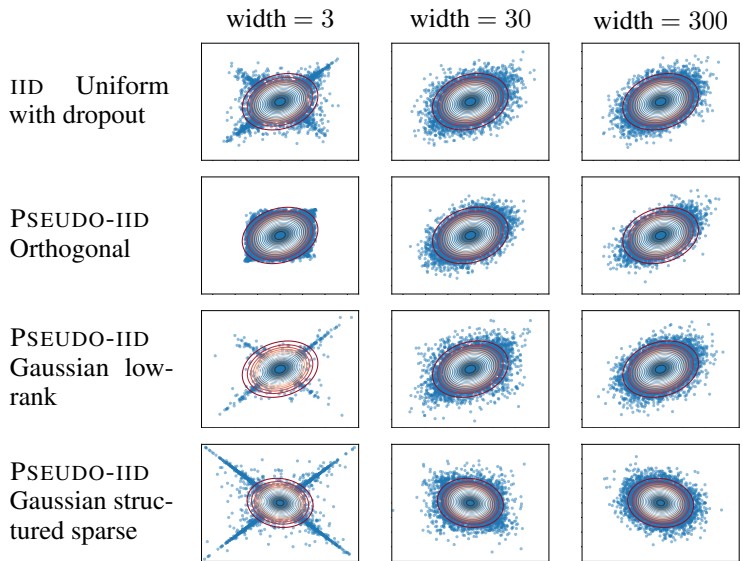

Figure 3: The empirical joint distribution of the preactivations generated by two distinct inputs flowing through the network. The large width limiting distribution as defined in Theorem 1 is included as level curves. The input data $x_a, x_b$ are drawn IID from $\mathbb{S}^9$ and 10000 experiments were conducted on a 7-layer fully connected network. The horizontal and vertical axes in each subplot are respectively $h_1^{(5)}(x_a)$ and $h_1^{(5)}(x_b)$.

better accuracy compared to finite point-estimate neural networks Lee et al. (2017) and approximates finite Bayesian networks very well (Matthews et al., 2018, Section 5).

**Edge of Chaos (EoC).** Random initialization of deep neural networks plays an important role in their trainability by regulating how quantities like the variance of preactivations and the pairwise correlation between input signals propagate through layers. An inappropriate initialization causes the network to collapse to a constant function or become a chaotic mapping oversensitive to input perturbations. The Edge of Chaos (EoC) initialization strategy rectifies these issues, additionally causing the network to have gradients of consistent magnitude across layers. Calculation of the EoC requires integration with respect to the distribution of preactivations in intermediary layers, which, in general, is intractable. However, the Gaussian Process limit simplifies the EoC analysis as carried out in Poole et al. (2016) and Xiao et al. (2018). Moreover, under the same distributional assumption, Schoenholz et al. (2017); Pennington et al. (2018) show that initialization on the EoC achieves *dynamical isometry*, making backpropagation of errors stable. Our main contributions of Theorems 1 and 2 allow similar calculations to be made for PSEUDO-IID networks such as the examples of low-rank, structured sparse, and orthogonal CNN as shown in Section 3.1. The EoC for low-rank networks has been calculated in Nait Saada & Tanner (2023) under the assumption of Theorems 1 and 2 which was at that time unproven. Interestingly, in terms of signal propagation, structured sparse or low-rank initializations are equivalent to their dense and full-rank counterparts up to a rescaling of the variance by a fractional factor of sparsity or rank.

## 4  CONCLUSION

We proved in this paper a new Gaussian Process limit for deep neural networks initialized with possibly inter-dependent entries. Examples include orthogonal, low-rank and structured sparse networks which are particularly of interest due to their efficient implementation and their empirically observed enhanced accuracy. Our result makes possible exact Bayesian inference as well as tractable Edge of Chaos analysis, for a broader class of either fully connected or convolutional networks. We expect the present work paves the way for a better understanding of the training dynamics of the emerging deep neural networks with parsimonious representations.

## ACKNOWLEDGMENTS

We thank Juba Nait Saada for insightful discussion and feedback on the manuscript, as well as Alex Cliffe for valuable comments and help in drawing Figure 1. Thiziri Nait Saada is supported by the UK Engineering and Physical Sciences Research Council (EPSRC) through the grant EP/W523781/1. Jared Tanner is supported by the Hong Kong Innovation and Technology Commission (InnoHK Project CIMDA).

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

## A  Proof of Theorem 1: Gaussian Process behaviour in Fully Connected Networks in the Pseudo-iid regime

### A.1  Step 1: Reduction of the problem from countably-infinite to finite dimensional

Firstly, we must clarify in what sense a sequence of stochastic Processes $h_i[n], i \in \mathbb{N}$ converges *in distribution* to its limit $h_i[*]$. For a sequence of real-valued random variables, we can define convergence in distribution $X_n \xrightarrow{d} X$ by the following condition: $\mathbb{E}f(X_n) \to \mathbb{E}f(X)$ for all *continuous* functions $f : \mathbb{R} \to \mathbb{R}$. Similarly, we can define weak convergence for random objects taking values in $\mathbb{R}^{\mathbb{N}}$ (countably-indexed stochastic Processes), provided that we equip $\mathbb{R}^{\mathbb{N}}$ with a "good" topology[3] that maintains a sufficiently rich class of continuous functions. The metric $\rho$ on the space of real sequences $\mathbb{R}^{\mathbb{N}}$ defined by

$$\rho(h, h') := \sum_{i=1}^{\infty} 2^{-i} \min(1, |h_i - h'_i|) \tag{10}$$

is one example. Thus, we can speak of the weak convergence $h_i[n] \xrightarrow{d} h_i[*]$ in the sense that $\mathbb{E}f(h_i[n]) \to \mathbb{E}f(h_i[*])$ for all $f : \mathbb{R}^{\mathbb{N}} \to \mathbb{R}$ continuous with respect to the metric $\rho$.

Fortunately, to prove the weak convergence of infinite-dimensional distributions, it is sufficient to show the convergence of their finite-dimensional marginals Billingsley (1999).

### A.2  Step 2: Reduction of the problem from multidimensional to one-dimensional

Let $\mathcal{L} = \{(i_1, x_1), \cdots, (i_P, x_P)\}$ be a finite subset of the index set $[N_\ell] \times \mathcal{X}$. We need to show that the vector $\left(h_{i_p}^{(\ell)}(x_p)[n]\right) \in \mathbb{R}^P$ converges in distribution to $\left(h_{i_p}^{(\ell)}(x_p)[*]\right) \in \mathbb{R}^P$. By the Cramér-Wold theorem (Cramér & Wold (1936)), we may equivalently show the weak convergence of an arbitrary linear projection

$$\mathcal{T}^{(\ell)}(\alpha, \mathcal{L})[n] = \sum_{(i,x) \in \mathcal{L}} \alpha_{(i,x)} \left(h_i^{(\ell)}(x)[n] - b_i^{(\ell)}\right) \tag{11}$$

$$= \sum_{(i,x) \in \mathcal{L}} \sum_{j=1}^{N_{\ell-1}[n]} \alpha_{(i,x)} W_{ij}^{(\ell)} z_j^{(\ell-1)}(x)[n] \tag{12}$$

$$= \frac{1}{\sqrt{N_{\ell-1}[n]}} \sum_{j=1}^{N_{\ell-1}[n]} \gamma_j^{(\ell)}(\alpha, \mathcal{L})[n], \tag{13}$$

where

$$\gamma_j^{(\ell)}(\alpha, \mathcal{L})[n] := \sigma_W \sum_{(i,x) \in \mathcal{L}} \alpha_{(i,x)} \epsilon_{ij}^{(\ell)} z_j^{(\ell-1)}(x)[n], \tag{14}$$

and $\epsilon_{ij}^{(\ell)}$ is centered and normalized, i.e. $\mathbb{E}\epsilon_{ij}^{(\ell)} = 0$, $\mathbb{E}\epsilon_{ij}^{(\ell)2} = 1$. We will be using a suitable version of Central Limit Theorem (CLT) to prove the weak convergence of the series in equation 13 to a Gaussian random variable.

### A.3  Step 3: Use of an exchangeable Central Limit Theorem

The classical Central Limit Theorem (CLT) establishes that for a sequence of IID random variables, the properly scaled sample mean converges to a Gaussian random variable in distribution. Here we recall an extension of the Central Limit Theorem introduced in Blum & Rosenblatt (1956), where the independence assumption on the summands is relaxed and replaced by an exchangeability condition. The following statement is an adapted version derived in Matthews et al. (2018), which is more suited to our case.

---

[3]In fact, it needs to be a Polish space, i.e. a complete separable metric space.

**Theorem 3** (Matthews et al. (2018), Lemma 10). *For each positive integer $n$, let $(X_{n,j}; j \in \mathbb{N}^*)$ be an infinitely exchangeable Process with mean zero, finite variance $\sigma_n^2$, and finite absolute third moment. Suppose also that the variance has a limit $\lim_{n\to\infty} \sigma_n^2 = \sigma_*^2$. Define*

$$S_n := \frac{1}{\sqrt{N[n]}} \sum_{j=1}^{N[n]} X_{n,j}, \tag{15}$$

*where $N : \mathbb{N} \to \mathbb{N}$ is a strictly increasing function. If*

*(a)* $\mathbb{E}[X_{n,1} X_{n,2}] = 0$,

*(b)* $\lim_{n\to\infty} \mathbb{E}[X_{n,1}^2 X_{n,2}^2] = \sigma_*^4$,

*(c)* $\mathbb{E}[|X_{n,1}|^3] = o_{n\to\infty}(\sqrt{N[n]})$,

*then $S_n$ converges in distribution to $\mathcal{N}(0, \sigma_*^2)$.*

Comparing equation 13 with equation 15, we need to check if the summands $X_{n,j} := \gamma_j^{(\ell)}(\alpha, \mathcal{L})[n]$ satisfy the conditions of Theorem 3. We will carefully verify each condition in the following sections. At each step, we make a note in italics whenever loosening the assumption on the first layer's weights from being IID Gaussian to row-IID introduces a slight variation in the proof.

### A.3.1 EXCHANGEABILITY OF THE SUMMANDS

To apply Theorem 3, we must first prove that the random variables $\gamma_j^{(\ell)}(\alpha, \mathcal{L})[n]$ are exchangeable. Let us expand the expression for $\gamma_j^{(\ell)}(\alpha, \mathcal{L})[n]$ further and write it in terms of the preactivations of layer $\ell - 2$, i.e.

$$\gamma_j^{(\ell)}(\alpha, \mathcal{L})[n] = \sigma_W \sum_{(i,x)\in\mathcal{L}} \alpha_{(i,x)} \epsilon_{ij}^{(\ell)} \phi(h_j^{(\ell-1)}(x)[n])$$

$$= \sigma_W \sum_{(i,x)\in\mathcal{L}} \alpha_{(i,x)} \epsilon_{ij}^{(\ell)} \phi\Big( \sum_{k=1}^{N_{\ell-2}[n]} W_{jk}^{(\ell-1)} z_k^{(\ell-2)}(x)[n] + b_j^{(\ell-1)} \Big).$$

If we apply a random permutation on the index $j$, let us say $\sigma : \{1, \cdots, N_{\ell-1}\} \to \{1, \cdots, N_{\ell-1}\}$, then the row-exchangeability and column-exchangeability of the PSEUDO-IID weights ensure that $W_{i\sigma(j)}^{(\ell)}$ has same distribution as $W_{ij}^{(\ell)}$ and that $W_{\sigma(j)k}^{(\ell-1)}$ has same distribution as $W_{jk}^{(\ell-1)}$, for any $i \in \{1, \cdots, N_\ell\}$, $k \in \{1, \cdots, N_{\ell-2}\}$. Note that this extends easily to the normalized versions of the weights $\epsilon_{ij}^{(\ell)}$. Additionally, as the biases are set to be IID Gaussians, they are a fortiori exchangeable and their distributions remain unchanged when considering random permutations of indices. Therefore, $\gamma_j^{(\ell)}(\alpha, \mathcal{L})[n]$ is equal to $\gamma_{\sigma(j)}^{(\ell)}(\alpha, \mathcal{L})[n]$ in distribution, hence exchangeability.

*For $\ell = 2$, when the weights $W_{jk}^{(1)}$ are row-IID only, they are also row-exchangeable and the exchangeability of $\gamma_j^{(2)}(\alpha, \mathcal{L})[n]$ follows.*

### A.3.2 MOMENT CONDITIONS

We mean by moment conditions the existence of a limiting variance $\sigma_*^2$, as well as the conditions (a)–(c) in Theorem 3. We will prove the moment conditions by induction on the layer number $\ell$.

**Existence of the limiting variance of the summands.** To show the existence of the limiting variance of the summands, let us first write down such a variance at a finite width. Since $\gamma_j^{(\ell)}$'s are exchangeable (see A.3.1), their distribution is identical and we may simply calculate the variance of

$\gamma_1^{(\ell)}$ as follows.

$$
\begin{aligned}
(\sigma^{(\ell)})^2(\alpha, \mathcal{L})[n] &:= \mathbb{E}\big(\gamma_1^{(\ell)}(\alpha, \mathcal{L})[n]\big)^2 = \mathbb{E}\Big[\sigma_W \sum_{(i,x)\in\mathcal{L}} \alpha_{(i,x)} \epsilon_{i,1}^{(\ell)} z_1^{(\ell-1)}(x)[n]\Big]^2 \\
&= \sigma_W^2 \sum_{\substack{(i_a,x_a)\in\mathcal{L} \\ (i_b,x_b)\in\mathcal{L}}} \alpha_{(i_a,x_a)}\alpha_{(i_b,x_b)} \mathbb{E}\Big[\epsilon_{i_a,1}^{(\ell)}\epsilon_{i_b,1}^{(\ell)}\Big] \mathbb{E}\Big[z_1^{(\ell-1)}(x_a)[n] \cdot z_1^{(\ell-1)}(x_b)[n]\Big] \\
&= \sigma_W^2 \sum_{\substack{(i_a,x_a)\in\mathcal{L} \\ (i_b,x_b)\in\mathcal{L}}} \alpha_{(i_a,x_a)}\alpha_{(i_b,x_b)} \delta_{i_a,i_b} \mathbb{E}\Big[z_1^{(\ell-1)}(x_a)[n] \cdot z_1^{(\ell-1)}(x_b)[n]\Big], \qquad (16)
\end{aligned}
$$

where we first considered the independence between the normalized weights at layer $\ell$ and the activations at layer $\ell - 1$; then used the fact that the normalized weights are uncorrelated in the PSEUDO-IID regime.

The convergence of the second moment of the summands is thus dictated by the convergence of the covariance of the activations of the last layer. By the induction hypothesis, the feature maps $h_j^{(\ell-1)}(x)[n]$ converge in distribution, so the continuous mapping theorem guarantees the existence of a limiting distribution for the activations $z_1^{(\ell-1)}(x_a)[n]$ and $z_1^{(\ell-1)}(x_b)[n]$ as $n$ tends to infinity. Note that this result holds even if the activation function has a set of discontinuity points of Lebesgue measure zero, e.g. the step function. Thus, the product inside the expectation in equation 16 converges in distribution to a limiting random variable $z_1^{(\ell-1)}(x_a)[*]z_1^{(\ell-1)}(x_b)[*]$.

From Billingsley (1999), one knows that if a sequence weakly converges to a limiting distribution and is uniformly integrable, then we can swap the order of taking the limit and the expectation. Thus, by Proposition 1, we have

$$
\begin{aligned}
(\sigma^{(\ell)})^2(\alpha, \mathcal{L})[*] &:= \lim_{n\to\infty} (\sigma^{(\ell)})^2(\alpha, \mathcal{L})[n] \\
&= \sigma_W^2 \sum_{\substack{(i_a,x_a)\in\mathcal{L} \\ (i_b,x_b)\in\mathcal{L}}} \alpha_{(i_a,x_a)}\alpha_{(i_b,x_b)} \delta_{i_a,i_b} \mathbb{E}\Big[z_1^{(\ell-1)}(x_a)[*] \cdot z_1^{(\ell-1)}(x_b)[*]\Big]. \qquad (17)
\end{aligned}
$$

*For $\ell = 2$, the activation $z_1^{(1)}(x_b)$ does not scale with n, i.e. $z_1^{(1)}(x_b) := z_1^{(1)}(x_b)[*]$, thus swapping expectation and limit is trivial.*

**Condition (a).** At a given layer $\ell$, we need to show that $X_{n,1} := \gamma_1^{(\ell)}(\alpha, \mathcal{L})[n]$ and $X_{n,2} := \gamma_2^{(\ell)}(\alpha, \mathcal{L})[n]$ are uncorrelated. We have

$$
\begin{aligned}
\mathbb{E}(X_{n,1}X_{n,2}) &= \mathbb{E}\Big[\Big(\sigma_W \sum_{(i,x)\in\mathcal{L}} \alpha_{(i,x)}\epsilon_{i,1}^{(\ell)} z_1^{(\ell-1)}(x)[n]\Big) \cdot \Big(\sigma_W \sum_{(i,x)\in\mathcal{L}} \alpha_{(i,x)}\epsilon_{i,2}^{(\ell)} z_2^{(\ell-1)}(x)[n]\Big)\Big] \\
&= \sigma_W^2 \sum_{\substack{(i_a,x_a)\in\mathcal{L} \\ (i_b,x_b)\in\mathcal{L}}} \alpha_{(i_a,x_a)}\alpha_{(i_b,x_b)} \mathbb{E}\Big[\epsilon_{i_a,1}^{(\ell)}\epsilon_{i_b,2}^{(\ell)}\Big] \mathbb{E}\Big[z_1^{(\ell-1)}(x_a)[n] \cdot z_2^{(\ell-1)}(x_b)[n]\Big] \\
&= 0,
\end{aligned}
$$

since $\epsilon_{i_a,1}^{(\ell)}$ and $\epsilon_{i_b,2}^{(\ell)}$ are uncorrelated by the PSEUDO-IID assumption, for $\ell \geq 2$.

**Condition (b).**

$$
\begin{aligned}
\mathbb{E}\Big[X_{n,1}^2 X_{n,2}^2\Big] &= \mathbb{E}\Big[\Big(\sigma_W \sum_{(i,x)\in\mathcal{L}} \alpha_{(i,x)} \epsilon_{i,1}^{(\ell)} z_1^{(\ell-1)}(x)[n]\Big)^2 \cdot \Big(\sigma_W \sum_{(i,x)\in\mathcal{L}} \alpha_{(i,x)} \epsilon_{i,2}^{(\ell)} z_2^{(\ell-1)}(x)[n]\Big)^2\Big] \\
&= \sigma_W^4 \sum_{\substack{(i_t,x_t)\in\mathcal{L} \\ t\in\{a,b,c,d\}}} \Big(\prod_{t\in\{a,b,c,d\}} \alpha_{(i_t,x_t)}\Big) \\
&\qquad\qquad \mathbb{E}\Big[\Big(\prod_{t\in\{a,b\}} \epsilon_{i_t,1}^{(\ell)} z_1^{(\ell-1)}(x_t)[n]\Big) \cdot \Big(\prod_{t\in\{c,d\}} \epsilon_{i_t,2}^{(\ell)} z_2^{(\ell-1)}(x_t)[n]\Big)\Big] \\
&= \sigma_W^4 \sum_{\substack{(i_t,x_t)\in\mathcal{L} \\ t\in\{a,b,c,d\}}} \Big(\prod_{t\in\{a,b,c,d\}} \alpha_{(i_t,x_t)}\Big) F_n(i_a,i_b,i_c,i_d) \\
&\qquad\qquad \mathbb{E}\Big[\Big(\prod_{t\in\{a,b\}} z_1^{(\ell-1)}(x_t)[n]\Big) \cdot \Big(\prod_{t\in\{c,d\}} z_2^{(\ell-1)}(x_t)[n]\Big)\Big],
\end{aligned}
\tag{18}
$$

where

$$
F_n(i_a,i_b,i_c,i_d) := \mathbb{E}\big[\epsilon_{i_a,1}^{(\ell)} \epsilon_{i_b,1}^{(\ell)} \epsilon_{i_c,2}^{(\ell)} \epsilon_{i_d,2}^{(\ell)}\big].
\tag{19}
$$

We justify the convergence in distribution of the random variable inside the expectation in the exact same way as in the previous section, referring to the continuous mapping theorem and the induction hypothesis. By Proposition 1, as $n \to \infty$, the above expectation converges to the expectation of the limiting Gaussian Process, i.e.

$$
\begin{aligned}
\lim_{n\to\infty} \mathbb{E}\Big[\Big(&\prod_{t\in\{a,b\}} z_1^{(\ell-1)}(x_t)[n]\Big) \cdot \Big(\prod_{t\in\{c,d\}} z_2^{(\ell-1)}(x_t)[n]\Big)\Big] \\
&= \mathbb{E}\Big[\Big(\prod_{t\in\{a,b\}} z_1^{(\ell-1)}(x_t)[*]\Big) \cdot \Big(\prod_{t\in\{c,d\}} z_2^{(\ell-1)}(x_t)[*]\Big)\Big].
\end{aligned}
$$

Moreover, condition (iv) of Definition 2 implies that

$$
\lim_{n\to\infty} F_n(i_a,i_b,i_c,i_d) = \delta_{i_a,i_b} \delta_{i_c,i_d}.
$$

Substituting the two limits back in the equation 18 and using the independence of the activations at layer $\ell - 1$ given by the induction hypothesis, we get

$$
\begin{aligned}
\lim_{n\to\infty} \mathbb{E}\Big[X_{n,1}^2 X_{n,2}^2\Big] &= \sigma_W^4 \sum_{\substack{(i_t,x_t)\in\mathcal{L} \\ t\in\{a,b,c,d\}}} \Big(\prod_{t\in\{a,b,c,d\}} \alpha_{(i_t,x_t)}\Big) \delta_{i_a,i_b} \delta_{i_c,i_d} \\
&\qquad\qquad \mathbb{E}\Big[\Big(\prod_{t\in\{a,b\}} z_1^{(\ell-1)}(x_t)[*]\Big) \cdot \Big(\prod_{t\in\{c,d\}} z_2^{(\ell-1)}(x_t)[*]\Big)\Big] \\
&= \sigma_W^4 \Big(\sum_{\substack{(i_t,x_t)\in\mathcal{L} \\ t\in\{a,b\}}} \Big(\prod_{t\in\{a,b\}} \alpha_{(i_t,x_t)}\Big) \delta_{i_a,i_b} \mathbb{E}\Big[\prod_{t\in\{a,b\}} z_1^{(\ell-1)}(x_t)[*]\Big]\Big) \\
&\qquad\qquad \Big(\sum_{\substack{(i_t,x_t)\in\mathcal{L} \\ t\in\{c,d\}}} \Big(\prod_{t\in\{c,d\}} \alpha_{(i_t,x_t)}\Big) \delta_{i_c,i_d} \mathbb{E}\Big[\prod_{t\in\{c,d\}} z_1^{(\ell-1)}(x_t)[*]\Big]\Big) \\
&= \sigma_W^4 (\sigma^{(\ell)})^4 (\alpha,\mathcal{L})[*].
\end{aligned}
$$

*For $\ell = 2$, the activations $z_1^{(1)}(x_t)$ do not scale with $n$, i.e. $z_1^{(1)}(x_t) := z_1^{(1)}(x_t)[*]$, thus swapping expectation and limit is trivial once again. The independence between the activations at different neurons boils down to the independence between the rows of $W^{(\ell)}$.*

**Condition (c).** To show that the third absolute moment of the $\gamma_1^{(\ell)}(\alpha, \mathcal{L})[n]$ grows slower than $\sqrt{N_{\ell-1}[n]}$ as $n \to \infty$, it is sufficient to bound it by a constant. Applying Hölder's inequality on $X = \left|\gamma_1^{(\ell)}(\alpha, \mathcal{L})[n]\right|^3, Y = 1, p = 4/3, q = 4$, we obtain

$$\mathbb{E}(\left|\gamma_1^{(\ell)}(\alpha, \mathcal{L})[n]\right|^3) \leq \left[\mathbb{E}\left|\gamma_1^{(\ell)}(\alpha, \mathcal{L})[n]\right|^4\right]^{\frac{1}{4}} \times 1.$$

Therefore, condition (c) boils down to showing that $\lim_{n \to \infty} \mathbb{E}\left|\gamma_1^{(\ell)}(\alpha, \mathcal{L})[n]\right|^4$ is finite, for which we will once again make use of the uniform integrability of the feature maps derived in C. Define

$$G_n(i_a, i_b, i_c, i_d) := \mathbb{E}\left[\epsilon_{i_a,1}\epsilon_{i_b,1}\epsilon_{i_c,1}\epsilon_{i_d,1}\right], \tag{20}$$

and observe that

$$\mathbb{E}\left|\gamma_1^{(\ell)}(\alpha, \mathcal{L})[n]\right|^4 = \mathbb{E}\left[\left(\sigma_W \sum_{(i,x) \in \mathcal{L}} \alpha_{(i,x)} \epsilon_{i,1}^{(\ell)} z_1^{(\ell-1)}(x)[n]\right)^4\right]$$

$$= \sigma_W^4 \sum_{\substack{(i_t, x_t) \in \mathcal{L} \\ t \in \{a,b,c,d\}}} \left(\prod_{t \in \{a,b,c,d\}} \alpha_{(i_t, x_t)}\right) G_n(i_a, i_b, i_c, i_d) \mathbb{E}\left[\prod_{t \in \{a,b,c,d\}} z_1^{(\ell-1)}(x_t)[n]\right].$$

Using Cauchy-Schwarz inequality, we can bound $G_n$ by the fourth moment of the normalized weights, i.e.

$$G_n \leq \sqrt{\mathrm{Var}(\epsilon^{(\ell)}{}_{i_a,1}\epsilon^{(\ell)}{}_{i_b,1})\mathrm{Var}(\epsilon^{(\ell)}{}_{i_c,1}\epsilon^{(\ell)}{}_{i_d,1})} = \mathrm{Var}(\epsilon^{(\ell)}{}_{i_a,1}\epsilon^{(\ell)}{}_{i_b,1})$$

$$= \mathbb{E}\left[\epsilon^{(\ell)}{}_{i_a,1}^2 \epsilon^{(\ell)}{}_{i_b,1}^2\right] \leq \mathrm{Var}(\epsilon^{(\ell)}{}_{i_a,1}^2) = \mathbb{E}\left[\epsilon^{(\ell)}{}_{i_a,1}^4\right] - 1.$$

Then, we use condition (iii) of the Definition 2 for $\mathbf{a} = (1, 0, \cdots, 0)^\top$ combined with Hölder's inequality 1 for $p = 2$ to bound the fourth moment:

$$\mathbb{E}\left[\epsilon^{(\ell)}{}_{i_a,1}^4\right] = \frac{n^2}{\sigma^4}\mathbb{E}\left[W_{i_a,1}^4\right]$$

$$= \frac{n^2}{\sigma^4}K_4\|\mathbf{a}\|_2^4 n^{-2} = \frac{K_4}{\sigma^4} = o_n(1).$$

Furthermore, the induction hypothesis gives the convergence in distribution of the feature maps from the last layer, and combined with the continuous mapping theorem we get the convergence in distribution of the above product inside expectation. Using Lemma 1, the uniform integrability of the activations follows, and Billingsley's theorem (Lemma 2) enables us to swap the limit and the expectation. Thus,

$$\lim_{n \to \infty} \mathbb{E}\left[\prod_{t \in \{a,b,c,d\}} z_1^{(\ell-1)}(x_t)[n]\right] = \mathbb{E}\left[\prod_{t \in \{a,b,c,d\}} z_1^{(\ell-1)}(x_t)[*]\right].$$

*For $\ell = 2$, the activations $z_1^{(1)}(x_t)$ do not scale with n, i.e. $z_1^{(1)}(x_t) := z_1^{(1)}(x_t)[*]$, thus swapping expectation and limit is trivial once again.*

To bound the product of four different random variables on $\mathbb{R}$, it is sufficient to bound the fourth order moment of each (see Lemma 3). We can do so using the linear envelope property (Definition 3) satisfied by the activation function to get, for any $(i, x) \in \mathcal{L}$,

$$\mathbb{E}\left[z_1^{(\ell-1)}(x)[*]^4\right] \leq 2^{4-1}\mathbb{E}\left[c^4 + M^4\left|h_1^{(\ell-1)}(x)[*]\right|^4\right].$$

The induction hypothesis indicates that $h_1^{(\ell-1)}(x)[*]$ follows a Gaussian distribution, whose fourth moment is bounded. Using the fact that we chose the set $\mathcal{L}$ to be finite, we can take the supremum over all $x$. Therefore, we have

$$\mathbb{E}\left[z_1^{(\ell-1)}(x_t)[*]^4\right] \leq 2^{4-1}\sup_{(i,x) \in \mathcal{L}} \mathbb{E}\left[c^4 + M^4\left|h_1^{(\ell-1)}(x)[*]\right|^4\right] = o_n(1).$$

Combining the above bounds, we then have

$$\lim_{n \to \infty} \mathbb{E}\big|\gamma_1^{(\ell)}(\alpha, \mathcal{L})[n]\big|^4 < \infty.$$

*For $\ell = 2$, the preactivation $h_1^{(1)}(x) = \sum_{j=1}^{N_0} W_{1j}^{(1)} x_j + b_1^{(1)}$ have a finite fourth moment provided that $W_{1j}^{(1)}$ does, which we assume in the row-*IID* case.*

### A.3.3 CONCLUSION FROM THE EXCHANGEABLE CLT

In the above sections, we showed by induction that at any depth, if the feature maps from previous layers converge in distribution to IID Gaussian Processes, then the assumptions of Theorem 3 hold and the one-dimensional projection of the feature maps at the current layer also converges in distribution to a Gaussian random variable with a specified variance. More precisely, we showed that for any finite set $\mathcal{L}$ and projection vector $\alpha$, any linear one-dimensional projection of the feature maps at the current layer, $\mathcal{T}^{(\ell)}(\alpha, \mathcal{L})[n]$, converges in distribution to a Gaussian $\mathcal{N}\big(0, \sigma^{(\ell)}(\alpha, \mathcal{L})[*]\big)$ as $n$ grows. This gives the convergence of the feature maps at layer $\ell$ to Gaussian Processes.

Considering the unbiased quantity

$$\mathcal{T}^{(\ell)}(\alpha, \mathcal{L})[*] := \sum_{(i,x) \in \mathcal{L}} \alpha_{(i,x)} \big(h_i^{(\ell)}(x)[*] - b_i^{(\ell)}\big),$$

we can compute its variance:

$$\mathbb{E}\big[\mathcal{T}^{(\ell)}(\alpha, \mathcal{L})[*]\big]^2 = \sum_{\substack{(i_a, x_a) \in \mathcal{L} \\ (i_b, x_b) \in \mathcal{L}}} \alpha_{(i_a, x_a)} \alpha_{(i_b, x_b)} \Big(\mathbb{E}\Big[h_{i_a}^{(\ell)}(x_a)[*] \cdot h_{i_b}^{(\ell)}(x_b)[*]\Big] - \sigma_b^2 \delta_{i_a, i_b}\Big).$$

As we saw in the previous sections,

$$(\sigma^{(\ell)})^2(\alpha, \mathcal{L})[*] = \sigma_W^2 \sum_{\substack{(i_a, x_a) \in \mathcal{L} \\ (i_b, x_b) \in \mathcal{L}}} \alpha_{(i_a, x_a)} \alpha_{(i_b, x_b)} \delta_{i_a, i_b} \mathbb{E}\Big[z_1^{(\ell-1)}(x_a)[*] \cdot z_1^{(\ell-1)}(x_b)[*]\Big].$$

Thus, by identification, and using the inductive hypothesis, one recovers the recursion formula for the variance as described in Theorem 1: For any $i_a, i_b \in \mathbb{N}$, $x_a, x_b \in \mathcal{X}$,

$$\mathbb{E}\Big[h_{i_a}^{(\ell)}(x_a)[*] h_{i_b}^{(\ell)}(x_b)[*]\Big] = \delta_{i_a, i_b}\Big(\sigma_W^2 \mathbb{E}\Big[z_1^{(\ell-1)}(x_a)[*] \cdot z_1^{(\ell-1)}(x_b)[*]\Big] + \sigma_b^2\Big)$$

$$= \delta_{i_a, i_b}\Big(\sigma_W^2 \mathbb{E}_{(u,v) \sim \mathcal{N}\big(\mathbf{0}, K^{(\ell-1)}(x, x')\big)} \big[\phi(u)\phi(v)\big] + \sigma_b^2\Big).$$

*For $\ell = 2$, if we denote by $\mathcal{D}$ the distribution of the weights at the first layer, we obtain instead*

$$\mathbb{E}\Big[h_{i_a}^{(2)}(x_a)[*] h_{i_b}^{(2)}(x_b)[*]\Big] = \delta_{i_a, i_b}\Big(\sigma_W^2 \mathbb{E}_{\mathcal{D}}\big[z_1^{(1)}(x_a) z_1^{(1)}(x_b)\big] + \sigma_b^2\Big). \tag{21}$$

### A.4 IDENTICAL DISTRIBUTION AND INDEPENDENCE OVER NEURONS

As we saw, for any $n$, the feature maps $h_j^{(\ell)}(\alpha, \mathcal{L})[n]$ are exchangeable, and, in particular, identically distributed. This still holds after taking the limit, that is $h_j^{(\ell)}[*]$ and $h_k^{(\ell)}[*]$ have the same distribution for any $j, k \in \mathbb{N}$.

It still remains to show the independence between $h_i^{(\ell)}[*]$ and $h_j^{(\ell)}[*]$ for $i \neq j$. As we now know the limiting distribution is Gaussian, it suffices to analyze their covariance to conclude about their independence. As derived in the previous section, for any $x, x' \in \mathcal{X}$, $i \neq j$, $\mathbb{E}\Big[h_i^{(\ell)}(x)[*] h_j^{(\ell)}(x')[*]\Big] = 0$, hence the independence.

## B    PROOF OF THEOREM 2: GAUSSIAN PROCESS BEHAVIOUR IN CONVOLUTIONAL NEURAL NETWORKS IN THE PSEUDO-IID REGIME

We apply the same machinery to show the Gaussian Process behaviour in CNNs under the PSEUDO-IID regime, closely following the steps detailed in the fully connected case. To reduce the problem to a simpler one, one can proceed as previously by considering a finite subset of the feature maps at layer $\ell$, $\mathcal{L} = \{(i_1, \mathbf{X}_1, \boldsymbol{\mu}_1), \cdots, (i_P, \mathbf{X}_P, \boldsymbol{\mu}_P)\} \subseteq [C_\ell] \times \mathcal{X} \times \mathcal{I}$, where $\mathcal{I}$ consists of all the spatial multi-indices. We will follow the same strategy outlined in Sections A.1-A.3. Given a finite set $\mathcal{L}$ and the projection vector $\alpha \in \mathbb{R}^{|\mathcal{L}|}$, we may form the unbiased one-dimensional projection as

$$\mathcal{T}^{(\ell)}(\alpha, \mathcal{L})[n] := \sum_{(i,\mathbf{X},\boldsymbol{\mu}) \in \mathcal{L}} \alpha_{(i,\mathbf{X},\boldsymbol{\mu})} \big(h_{i,\boldsymbol{\mu}}^{(\ell)}(\mathbf{X})[n] - b_i^{(\ell)}\big), \tag{22}$$

which can be rewritten, using equation 5, as the sum

$$\mathcal{T}^{(\ell)}(\alpha, \mathcal{L})[n] = \frac{1}{\sqrt{C_{\ell-1}[n]}} \sum_{j=1}^{C_{\ell-1}[n]} \gamma_j^{(\ell)}(\alpha, \mathcal{L})[n], \tag{23}$$

where the summands are

$$\gamma_j^{(\ell)}(\alpha, \mathcal{L})[n] := \sigma_W \sum_{(i,\mathbf{X},\boldsymbol{\mu}) \in \mathcal{L}} \alpha_{(i,\mathbf{X},\boldsymbol{\mu})} \sum_{\boldsymbol{\nu} \in [\![\boldsymbol{\mu}]\!]} \mathbf{E}_{i,j,\boldsymbol{\nu}}^{(\ell)} z_{j,\boldsymbol{\nu}}^{(\ell-1)}(\mathbf{X})[n]. \tag{24}$$

As before, we introduced the renormalized version $\mathbf{E}^{(\ell)}$ of the filter $\mathbf{U}^{(\ell)}$ such that $\mathbf{E}_{i,j,\boldsymbol{\nu}}^{(\ell)} := \sigma_W^{-1}\sqrt{C_\ell}\mathbf{U}_{i,j,\boldsymbol{\nu}}^{(\ell)}$.

We will proceed once again by induction forward through the network's layer verifying the assumptions of the exchangeable CLT (Theorem 3) and using it on $\mathcal{T}^{(\ell)}(\alpha, \mathcal{L})[n]$ to conclude its convergence in distribution to a Gaussian random variable.

**The exchangeability of the summands.**    Similar to the fully connected case, we employ the row- and column-exchangeability of the PSEUDO-IID convolutional kernel to show the exchangeability of $\gamma_j^{(\ell)}$'s. Let us expand equation 24 and write

$$\gamma_j^{(\ell)}(\alpha, \mathcal{L})[n] = \sigma_W \sum_{(i,\mathbf{X},\boldsymbol{\mu}) \in \mathcal{L}} \alpha_{(i,\mathbf{X},\boldsymbol{\mu})} \sum_{\boldsymbol{\nu} \in [\![\boldsymbol{\mu}]\!]} \mathbf{E}_{i,j,\boldsymbol{\nu}}^{(\ell)} \phi(h_{j,\boldsymbol{\nu}}^{(\ell-1)}(\mathbf{X})[n])$$

$$= \sigma_W \sum_{(i,\mathbf{X},\boldsymbol{\mu}) \in \mathcal{L}} \sum_{\boldsymbol{\nu} \in [\![\boldsymbol{\mu}]\!]} \alpha_{(i,\mathbf{X},\boldsymbol{\mu})} \mathbf{E}_{i,j,\boldsymbol{\nu}}^{(\ell)} \phi\Big(\sum_{k=1}^{C_{\ell-2}[n]} \sum_{\boldsymbol{\xi} \in [\![\boldsymbol{\nu}]\!]} \mathbf{U}_{j,k,\boldsymbol{\xi}}^{(\ell-1)} z_{k,\boldsymbol{\xi}}^{(\ell-2)}(\mathbf{X})[n]\Big).$$

The joint distributions of $\big(\mathbf{U}_{i,j,\boldsymbol{\nu}}^{(\ell)}\big)_{j=1}^{C_{\ell-1}}$ and $\big(\mathbf{U}_{j,k,\boldsymbol{\xi}}^{(\ell-1)}\big)_{j=1}^{C_{\ell-1}}$ are invariant under any permutation $j \mapsto \sigma(j)$, therefore $\gamma_j^{(\ell)}$'s are exchangeable.

**Moment conditions.**    Condition (a) is straightforward as the filters' entries are uncorrelated by the PSEUDO-IID assumption: $\mathbb{E}\big[\mathbf{U}_{i,j,\boldsymbol{\mu}}^{(\ell)} \mathbf{U}_{i',j',\boldsymbol{\mu}'}^{(\ell)}\big] = \frac{\sigma_W^2}{C_{\ell-1}} \delta_{i,i'} \delta_{j,j'} \delta_{\boldsymbol{\mu},\boldsymbol{\mu}'}$.

The moment conditions are shown to be satisfied by induction through the network and the proofs boil down to showing the uniform integrability of the activation vectors to be able to swap limit and expectation. This uniform integrability in the CNN case under PSEUDO-IID weights is rigorously demonstrated in Proposition 2. One can compute the variance of one representative of the summands, let us say the first one, as follows:

$$(\sigma^{(\ell)})^2(\alpha,\mathcal{L})[n] := \mathbb{E}\big(\gamma_1^{(\ell)}(\alpha,\mathcal{L})[n]^2\big) = \mathbb{E}\Big[\sigma_W \sum_{(i,\mathbf{X},\boldsymbol{\mu})\in\mathcal{L}} \alpha_{(i,\mathbf{X},\boldsymbol{\mu})} \sum_{\boldsymbol{\nu}\in[\![\boldsymbol{\mu}]\!]} \mathbf{E}_{i,1,\boldsymbol{\nu}}^{(\ell)} z_{1,\boldsymbol{\nu}}^{(\ell-1)}(\mathbf{X})[n]\Big]^2$$

$$= \sigma_W^2 \sum_{\substack{(i_a,\mathbf{X}_a,\boldsymbol{\mu}_a)\in\mathcal{L}\\(i_b,\mathbf{X}_b,\boldsymbol{\mu}_b)\in\mathcal{L}}} \alpha_{(i_a,\mathbf{X}_a,\boldsymbol{\mu}_a)}\alpha_{(i_b,\mathbf{X}_b,\boldsymbol{\mu}_b)}$$

$$\sum_{\substack{\boldsymbol{\nu}_a\in[\![\boldsymbol{\mu}_a]\!]\\\boldsymbol{\nu}_b\in[\![\boldsymbol{\mu}_b]\!]}} \mathbb{E}\Big[\mathbf{E}_{i_a,1,\boldsymbol{\nu}_a}^{(\ell)}\mathbf{E}_{i_b,1,\boldsymbol{\nu}_b}^{(\ell)}\Big]\mathbb{E}\Big[z_{1,\boldsymbol{\nu}_a}^{(\ell-1)}(\mathbf{X}_a)[n]\cdot z_{1,\boldsymbol{\nu}_b}^{(\ell-1)}(\mathbf{X}_b)[n]\Big]$$

$$= \sigma_W^2 \sum_{\substack{(i_a,\mathbf{X}_a,\boldsymbol{\mu}_a)\in\mathcal{L}\\(i_b,\mathbf{X}_b,\boldsymbol{\mu}_b)\in\mathcal{L}}} \alpha_{(i_a,\mathbf{X}_a,\boldsymbol{\mu}_a)}\alpha_{(i_b,\mathbf{X}_b,\boldsymbol{\mu}_b)}$$

$$\sum_{\substack{\boldsymbol{\nu}_a\in[\![\boldsymbol{\mu}_a]\!]\\\boldsymbol{\nu}_b\in[\![\boldsymbol{\mu}_b]\!]}} \mathbb{E}\Big[\mathbf{E}_{i_a,1,\boldsymbol{\nu}_a}^{(\ell)}\mathbf{E}_{i_b,1,\boldsymbol{\nu}_b}^{(\ell)}\Big]\mathbb{E}\Big[z_{1,\boldsymbol{\nu}_a}^{(\ell-1)}(\mathbf{X}_a)[n]\cdot z_{1,\boldsymbol{\nu}_b}^{(\ell-1)}(\mathbf{X}_b)[n]\Big]$$

$$= \sigma_W^2 \sum_{\substack{(i_a,\mathbf{X}_a,\boldsymbol{\mu}_a)\in\mathcal{L}\\(i_b,\mathbf{X}_b,\boldsymbol{\mu}_b)\in\mathcal{L}}} \alpha_{(i_a,\mathbf{X}_a,\boldsymbol{\mu}_a)}\alpha_{(i_b,\mathbf{X}_b,\boldsymbol{\mu}_b)}$$

$$\sum_{\substack{\boldsymbol{\nu}_a\in[\![\boldsymbol{\mu}_a]\!]\\\boldsymbol{\nu}_b\in[\![\boldsymbol{\mu}_b]\!]}} \delta_{i_a,i_b}\delta_{\boldsymbol{\nu}_a,\boldsymbol{\nu}_b}\mathbb{E}\Big[z_{1,\boldsymbol{\nu}_a}^{(\ell-1)}(\mathbf{X}_a)[n]\cdot z_{1,\boldsymbol{\nu}_b}^{(\ell-1)}(\mathbf{X}_b)[n]\Big].$$

Given the uniform integrability of the product the activations in a CNN, we may swap the order of taking the limit and the expectation to have

$$(\sigma^{(\ell)})^2(\alpha,\mathcal{L})[*] := \lim_{n\to\infty}(\sigma^{(\ell)})^2(\alpha,\mathcal{L})[n]$$

$$= \sigma_W^2 \sum_{\substack{(i_a,\mathbf{X}_a,\boldsymbol{\mu}_a)\in\mathcal{L}\\(i_b,\mathbf{X}_b,\boldsymbol{\mu}_b)\in\mathcal{L}}} \alpha_{(i_a,\mathbf{X}_a,\boldsymbol{\mu}_a)}\alpha_{(i_b,\mathbf{X}_b,\boldsymbol{\mu}_b)}$$

$$\sum_{\substack{\boldsymbol{\nu}_a\in[\![\boldsymbol{\mu}_a]\!]\\\boldsymbol{\nu}_b\in[\![\boldsymbol{\mu}_b]\!]}} \delta_{i_a,i_b}\delta_{\boldsymbol{\nu}_a,\boldsymbol{\nu}_b}\mathbb{E}\Big[z_{1,\boldsymbol{\nu}_a}^{(\ell-1)}(\mathbf{X}_a)[*]\cdot z_{1,\boldsymbol{\nu}_b}^{(\ell-1)}(\mathbf{X}_b)[*]\Big].$$

As in the fully connected case, we conclude that the feature maps at the next layer converge to Gaussians Processes, whose covariance function is given by

$$\mathbb{E}\Big[h_{i_a,\boldsymbol{\mu_a}}^{(\ell)}(\mathbf{X_a})[*]\cdot h_{i_b,\boldsymbol{\mu_b}}^{(\ell)}(\mathbf{X_b})[*]\Big] = \delta_{i_a,i_b}\Big(\sigma_b^2 + \sigma_W^2 \sum_{\boldsymbol{\nu}\in[\![\boldsymbol{\mu_a}]\!]\cap[\![\boldsymbol{\mu_b}]\!]} K_{\boldsymbol{\nu}}^{(\ell)}(\mathbf{X_a},\mathbf{X_b})\Big),$$

where

$$K_{\boldsymbol{\nu}}^{(\ell)}(\mathbf{X_a},\mathbf{X_b}) = \begin{cases} \mathbb{E}_{(u,v)\sim\mathcal{N}(\mathbf{0},K_{\boldsymbol{\nu}}^{(\ell-1)}(\mathbf{X_a},\mathbf{X_b}))}[\phi(u)\phi(v)], & \ell\geq 2\\ \frac{1}{C_0}\sum_{j=1}^{C_0}(\mathbf{X_a})_{j,\boldsymbol{\nu}}(\mathbf{X_b})_{j,\boldsymbol{\nu}}, & \ell=1 \end{cases}.$$

## C   Lemmas used in the proof of Theorems 1 and 2

We will present in this section the lemmas used in the derivation of our results. The proofs are omitted in cases where they can easily be found in the literature.

**Lemma 1** (Hölder's inequality)**.** *For a probability space* $(\Omega,\mathcal{F},\mathbb{P})$*, let* $X,Y$ *be two random variables on* $\Omega$ *and* $p,q>1$ *such that* $p^{-1}+q^{-1}=1$*. Then,*

$$\mathbb{E}|XY| \leq \big(\mathbb{E}|X|^p\big)^{\frac{1}{p}}\big(\mathbb{E}|Y|^q\big)^{\frac{1}{q}}.$$

**Lemma 2** (Bellingsley's theorem). *Let $X[n]$ be a sequence of random variables, for $n \in \mathbb{N}$ and $X$ another random variable. If $X[n]$ is uniformly integrable and the sequence $X[n]$ converges in distribution to $X[*] = X$, then $X$ is integrable and one can swap the limit and the expectation, i.e.*

$$\lim_{n \to \infty} \mathbb{E}X[n] = \mathbb{E}X[*].$$

We adapt Lemma 18 from Matthews et al. (2018) to our setting and the proof can trivially be obtained from the proof derived in the cited article.

**Lemma 3** (Sufficient condition to uniformly bound the expectation of a four-cross product). *Let $X_1$, $X_2$, $X_3$, and $X_4$ be random variables on $\mathbb{R}$ with the usual Borel $\sigma$-algebra. Assume that $\mathbb{E}|X_i|^4 < \infty$ for all $i \in \{1, 2, 3, 4\}$. Then, for any choice of $p_i \in \{0, 1, 2\}$ (where $i \in \{1, 2, 3, 4\}$), the expectations $\mathbb{E}\big[\prod_{i=1}^{4} |X_i|^{p_i}\big]$ are uniformly bounded by a polynomial in the eighth moments $\mathbb{E}|X_i|^8 < \infty$.*

This Lemma can be easily derived from standard Pearson correlation bounds.

We now have the tools to show that all four-cross products of the activations are uniformly integrable in the PSEUDO-IID setting, which enables us to swap the limit and the expectation in the previous sections of the appendix.

**Proposition 1** (Uniform integrability in the PSEUDO-IID regime – fully connected networks). *Consider a fully connected neural network in the PSEUDO-IID regime. Consider the random activations $z_i^{(\ell)}(x_a)[n], z_j^{(\ell)}(x_b)[n], z_k^{(\ell)}(x_c)[n], z_l^{(\ell)}(x_d)[n]$ with any $i, j, k, l \in \mathbb{N}, x_a, x_b, x_c, x_d \in \mathcal{X}$, neither necessarily distinct, as in equation 2. Then, the family of random variables*

$$z_i^{(\ell)}(x_a)[n]z_j^{(\ell)}(x_b)[n]z_k^{(\ell)}(x_c)[n]z_l^{(\ell)}(x_d)[n],$$

*indexed by $n$ is uniformly integrable for any $\ell = 1, \cdots L + 1$.*

*Proof.* The proof is adapted from Matthews et al. (2018) to the PSEUDO-IID regime in fully connected neural networks described in equation 2.

If a collection of random variables is uniformly $L^p$-bounded for $p > 1$, then it is uniformly integrable. So, we will show that our family of random variables is uniformly $L^{1+\epsilon}$-bounded for some $\epsilon > 0$, i.e. there exists $K < \infty$ independent of $n$ such that,

$$\mathbb{E}\Big|z_i^{(\ell)}(x_a)[n]z_j^{(\ell)}(x_b)[n]z_k^{(\ell)}(x_c)[n]z_l^{(\ell)}(x_d)[n]\Big|^{1+\epsilon} \leq K,$$

which is equivalent to

$$\mathbb{E}\Big[\big|z_i^{(\ell)}(x_a)[n]^{1+\epsilon}\big|\big|z_j^{(\ell)}(x_b)[n]^{1+\epsilon}\big|\big|z_k^{(\ell)}(x_c)[n]^{1+\epsilon}\big|\big|z_l^{(\ell)}(x_d)[n]^{1+\epsilon}\big|\Big] \leq K.$$

To do so, Lemma 3 gives us a sufficient condition: bounding the moment of order $4(1 + \epsilon)$ of each term in the product by a constant independent of $n$. For any $x_t \in \mathcal{X}$ and $i \in \mathbb{N}$, this moment can be rewritten in terms of the feature maps using the linear envelope property 3 and the convexity of the map $x \mapsto x^{4(1+\epsilon)}$ as

$$\mathbb{E}\Big[\big|z_i^{(\ell)}(x_t)[n]\big|^{4(1+\epsilon)}\Big] \leq 2^{4(1+\epsilon)-1}\mathbb{E}\Big[c^{4(1+\epsilon)} + M^{4(1+\epsilon)}\big|h_i^{(\ell)}(x_t)[n]\big|^{4(1+\epsilon)}\Big].$$

Thus it is sufficient to show that the absolute feature maps $\big|h_i^{(\ell)}(x_t)[n]\big|$ have a finite moment of order $4 + \epsilon$, independent of $x_t$, $i$, and $n$, for some $\epsilon$. For the sake of simplicity, we will show this by induction on $\ell$ for $\epsilon = 1$.

**Base case.** For $\ell = 1$, the feature maps $h_i^{(1)} = \sum_{j=1}^{N_0} W_{i,j}^{(1)}x_j + b_i^{(1)}$ are identically distributed for all $i \in \{1, \cdots, N_1[n]\}$ from the row-exchangeability of the weights. Moreover, from the moment condition of Definition 2, there exists $p = 8$ such that

$$\mathbb{E}\big|h_i^{(1)}(x)[n]\big|^p \leq \mathbb{E}\Big|\sum_{j=1}^{N_0} x_j W_{i,j}^{(1)}\Big|^p + \mathbb{E}\big|b_i^{(1)}\big|^p$$

$$= K\|\mathbf{x}\|_2^p N_0^{-p/2} + \mathbb{E}\big|b_i^{(1)}\big|^p.$$

The RHS of the above equality is independent of $N_1$ (and $n$). Therefore, for all $n \in \mathbb{N}$ and $i \in \{1, \cdots, N_1[n]\}$, we have found a constant bound for the feature map's moment of order $p = 8$.

**Inductive step.** Let us assume that for any $\{x_t\}_{t=1}^4$ and $i \in \mathbb{N}$, the eigth-order moment of $\left| h_i^{(\ell-1)}(x_t)[n] \right|$ is bounded by a constant independent from $n$.

We will show that this implies

$$\mathbb{E}\big[\big| h_i^{(\ell)}(x_t)[n]\big|^8\big] < \infty.$$

Considering the vector of activations $\phi\big(h_\odot^{(\ell-1)}(x_t)\big)[n]$, we have, from the moment condition (iii) of the PSEUDO-IID regime the following conditional expectation,

$$\mathbb{E}\Big[\Big| \sum_{j=1}^{N_{\ell-1}[n]} W_{ij}^{(\ell)}\phi\big(h_j^{(\ell-1)}(x_t)\big)[n] \Big|^8 \mid \phi\big(h_\odot^{(\ell-1)}(x_t)\big)[n]\Big] = K\|\phi\big(h_\odot^{(\ell-1)}(x_t)\big)[n]\|_2^8 N_{\ell-1}[n]^{-\frac{8}{2}}.$$

Thus using the recursion formulae for the data propagation in such architecture given by equation 2, the convexity of $x \mapsto x^8$ on $\mathbb{R}^+$ and taking conditional expectations,

$$\mathbb{E}\Big[\big| h_i^{(\ell)}(x_t)[n]\big|^8\Big] \leq 2^{8-1}\mathbb{E}\Big[\big|b_i^{(\ell)}\big|^8 + \Big| \sum_{j=1}^{N_{\ell-1}[n]} W_{ij}^{(\ell)}\phi\big(h_j^{(\ell-1)}(x_t)\big)[n] \Big|^8\Big] \tag{25}$$

$$= 2^{8-1}\mathbb{E}\Big[\big|b_i^{(\ell)}\big|^8\Big] + \mathbb{E}\Big[\mathbb{E}\Big| \sum_{j=1}^{N_{\ell-1}[n]} W_{ij}^{(\ell)}\phi\big(h_j^{(\ell-1)}(x_t)\big)[n] \Big|^8 \mid \phi\big(h_\odot^{(\ell-1)}(x_t)\big)[n]\Big]$$

$$= 2^{8-1}\mathbb{E}\Big[\big|b_i^{(\ell)}\big|^8\Big] + K N_{\ell-1}[n]^{-4}\mathbb{E}\Big[\|\phi\big(h_\odot^{(\ell-1)}(x_t)\big)[n]\|_2^8\Big]. \tag{26}$$

As the biases are IID Gaussians, the first expectation involving the bias is trivially finite and does not depend on $i$ as they are identically distributed. For the second expectation, we compute,

$$\mathbb{E}\|\phi\big(h_\odot^{(\ell-1)}(x_t)\big)[n]\|_2^8 = \mathbb{E}\Big[ \sum_{j=1}^{N_{\ell-1}[n]} \Big(\phi\big(h_j^{(\ell-1)}(x_t)\big)[n]\Big)^2\Big]^{\frac{8}{2}} \tag{27}$$

$$\leq \mathbb{E}\Big[ \sum_{j=1}^{N_{\ell-1}[n]} \Big(c + M\big(h_j^{(\ell-1)}(x_t)\big)[n]\Big)^2\Big]^{\frac{8}{2}}$$

$$= \mathbb{E}\Big[ \sum_{j=1}^{N_{\ell-1}[n]} c^2 + M^2\big(h_j^{(\ell-1)}(x_t)\big)^2[n] + 2cM\big(h_j^{(\ell-1)}(x_t)\big)[n]\Big]^4.$$

This last expression can be written as a linear combination of $N_{\ell-1}[n]^4$ quantities of the form

$$\mathbb{E}\Big[\big(h_{j_1}^{(\ell-1)}(x_t)\big)^{p_1}[n]\big(h_{j_2}^{(\ell-1)}(x_t)\big)^{p_2}[n]\big(h_{j_3}^{(\ell-1)}(x_t)\big)^{p_3}[n]\big(h_{j_4}^{(\ell-1)}(x_t)\big)^{p_4}[n]\Big],$$

and we bound each of them making use of Lemma 3. The factor $N_{\ell-1}[n]^{-4}$ in equation 26 thus cancels out with the number of terms in the sum equation 27. As the preactivations are exchangeable, they are identically distributed so the dependence on the neuron index $i$ can be ignored, and taking the supremum over the finite set of input data $\{x_t\}$ does not affect the uniformity of the bound; which concludes the proof.

$\square$

**Proposition 2** (Uniform integrability in the PSEUDO-IID regime – CNN.). *Consider a convolutional neural network in the PSEUDO-IID regime. Consider a collection of random variables*

$z_i^{(\ell)}(x_a)[n], z_j^{(\ell)}(x_b)[n], z_k^{(\ell)}(x_c)[n], z_l^{(\ell)}(x_d)[n]$ with any $i, j, k, l \in \mathbb{N}, \mathbf{X}_a, \mathbf{X}_b, \mathbf{X}_c, \mathbf{X}_d \in \mathcal{X}$, neither necessarily distinct, obtained by the recursion equation 5. Then, the family of random variables

$$z_i^{(\ell)}(\mathbf{X}_a)[n] z_j^{(\ell)}(\mathbf{X}_b)[n] z_k^{(\ell)}(\mathbf{X}_c)[n] z_l^{(\ell)}(\mathbf{X}_d)[n],$$

indexed by $n$ is uniformly integrable for any $\ell = \{1, \cdots L + 1\}$.

*Proof.* As previously, this proposition holds some novelty of this paper, extending already known proofs in the standard Gaussian IID setting to the PSEUDO-IID regime in CNNs. Note that this directly implies the universality of the Gaussian Process behaviour for CNNs in the IID regime, which has been established so far only by Yang (2021) to the best of our knowledge. Our result goes beyond. We recall that the data propagation is described by Equation equation 5.

Once again, we observe it is sufficient to show such that the moment of order 8 of the feature maps is uniformly bounded. We do this again by induction.

**Base case.** Since $h_{i,\boldsymbol{\mu}}^{(1)}(\mathbf{X})[n] = b_i^{(1)} + \sum_{j=1}^{C_0} \sum_{\boldsymbol{\nu} \in [\![\boldsymbol{\mu}]\!]} \mathbf{U}_{i,j,\boldsymbol{\nu}}^{(1)} x_{j,\boldsymbol{\nu}}$, and weights and biases are independent, for some $p = 8$ we can write

$$\mathbb{E} |h_{i,\boldsymbol{\mu}}^{(1)}(\mathbf{X})[n]|^p \leq \mathbb{E} \Big| \sum_{j=1}^{C_0} \sum_{\boldsymbol{\nu} \in [\![\boldsymbol{\mu}]\!]} x_{j,\boldsymbol{\nu}} \mathbf{U}_{i,j,\boldsymbol{\nu}}^{(1)} \Big|^p + \mathbb{E} |b_i^{(1)}|^p$$

$$= K_p \|\mathbf{X}_{[\![\boldsymbol{\mu}]\!]}\|_2^p C_0^{-p/2} + \mathbb{E} |b_i^{(1)}|^p,$$

where $\mathbf{X}_{[\![\boldsymbol{\mu}]\!]}$ is the part of signal around the pixel $\boldsymbol{\mu}$ and we have used the condition (iii) of Definition 4 in the last line. Observe that the RHS is independent of $i$ and $n$.

**Inductive step.** Let us assume that for any $\{\mathbf{X}_t\}_{t=1}^4$, $\boldsymbol{\mu} \in \mathcal{I}$ and $i \in \mathbb{N}$, there exists $\epsilon_0 \in (0, 1)$ such that the eighth moment of the preactivations from the previous layer $|h_{i,\boldsymbol{\mu}}^{(\ell-1)}(\mathbf{X}_t)[n]|$ is bounded by a constant independent from $j \in \mathbb{N}, \boldsymbol{\nu} \in \mathcal{I}$, and $n$, for all $\mathbf{X}_t \in \mathcal{X}$

Mirroring our proof in the fully connected case, we will show that this propagates to the next layer,

$$\mathbb{E} |h_{i,\boldsymbol{\mu}}^{(\ell)}(\mathbf{X}_t)[n]|^8 < \infty.$$

From the third condition of the PSEUDO-IID regime, we can compute the expectation conditioned on the vector of activations $\phi\big(h_{\odot,\odot}^{(\ell-1)}(\mathbf{X}_t)\big)[n]$,

$$\mathbb{E} \Big[ \Big| \sum_{j=1}^{C_{\ell-1}[n]} \sum_{\boldsymbol{\nu} \in [\![\boldsymbol{\mu}]\!]} \mathbf{U}_{i,j,\boldsymbol{\nu}}^{(\ell)} \phi\big(h_{j,\boldsymbol{\nu}}^{(\ell-1)}(\mathbf{X}_t)\big)[n] \Big|^8 \Big] = \mathbb{E} \Big[ \mathbb{E} \Big| \sum_{j=1}^{C_{\ell-1}[n]} \sum_{\boldsymbol{\nu} \in [\![\boldsymbol{\mu}]\!]} \mathbf{U}_{i,j,\boldsymbol{\nu}}^{(\ell)} \phi\big(h_{j,\boldsymbol{\nu}}^{(\ell-1)}(\mathbf{X}_t)\big)[n] \Big|^8$$

$$| \phi\big(h_{\odot,\odot}^{(\ell-1)}(\mathbf{X}_t)\big)[n] \Big]$$

$$= K C_{\ell-1}[n]^{-4} \mathbb{E} \|\phi\big(h_{\odot,\odot}^{(\ell-1)}(\mathbf{X}_t)\big)[n]\|_2^8,$$

where the norm of the activations can be computed using the linear envelope property,

$$\mathbb{E} \|\phi\big(h_{\odot,\odot}^{(\ell-1)}(\mathbf{X}_t)\big)[n]\|_2^8 \leq \mathbb{E} \Big[ \sum_{j=1}^{C_{\ell-1}[n]} \sum_{\boldsymbol{\nu} \in [\![\boldsymbol{\mu}]\!]} \big(\phi\big(h_{j,\boldsymbol{\nu}}^{(\ell-1)}(\mathbf{X}_t)\big)[n]\big)^2 \Big]^4$$

$$\leq \mathbb{E} \Big[ \sum_{j=1}^{C_{\ell-1}[n]} \sum_{\boldsymbol{\nu} \in [\![\boldsymbol{\mu}]\!]} c^2 + M^2 h_{j,\boldsymbol{\nu}}^{(\ell-1)}(\mathbf{X}_t)^2[n] + 2cM h_{j,\boldsymbol{\nu}}^{(\ell-1)}(\mathbf{X}_t) \Big]^4.$$

This last quantity turns out to be the weighted sum of $(k \times C_{\ell-1}[n])^4$ terms (recall $k$ being the filter size, which is finite) of the form

$$\mathbb{E}\Big[h_{j_1,\boldsymbol{\nu}}^{(\ell-1)}(\mathbf{X}_t)^{p_1}[n]h_{j_2,\boldsymbol{\nu}}^{(\ell-1)}(\mathbf{X}_t)^{p_2}[n]h_{j_3,\boldsymbol{\nu}}^{(\ell-1)}(\mathbf{X}_t)^{p_3}[n]h_{j_4,\boldsymbol{\nu}}^{(\ell-1)}(\mathbf{X}_t)^{p_4}[n]\Big],$$

that can be bounded using Lemma 3 combined with our inductive hypothesis. Observe how the factors $C_{\ell-1}[n]^4$ cancel out and lead to a bound independent from $n$.

Using the recursion formulae for the data propagation in the CNN architecture recalled in equation 5 and the convexity of the map $x \mapsto x^8$ on $\mathbb{R}^+$, we have

$$\mathbb{E}\big|h_{i,\boldsymbol{\mu}}^{(\ell)}(\mathbf{X}_t)[n]\big|^8 \leq 2^{8-1}\mathbb{E}\Big[\big|b_i^{(\ell)}\big|^8 + \Big|\sum_{j=1}^{C_{\ell-1}[n]}\sum_{\boldsymbol{\nu}\in[\![\boldsymbol{\mu}]\!]}\mathbf{U}_{i,j,\boldsymbol{\nu}}^{(\ell)}\phi\big(h_{j,\boldsymbol{\nu}}^{(\ell-1)}(\mathbf{X}_t)\big)[n]\Big|^8\Big].$$

The first expectation is finite and uniformly bounded as the biases are IID Gaussians in the PSEUDO-IID regime and we have just shown above the boundedness of the second expectation.

Therefore,

$$\mathbb{E}\big|\phi\big(h_{j,\boldsymbol{\nu}}^{(\ell-1)}(\mathbf{X}_t)\big)[n]\big|^8 < \infty, \tag{28}$$

and taking the supremum over the finite input data set does not change the uniformly bounded property, which was needed to be shown.

$\square$

## D    IID AND ORTHOGONAL WEIGHTS ARE PSEUDO-IID

**IID weights.** If $A = (A_{ij}) \in \mathbb{R}^{m \times n}$ has IID entries $A_{ij} \overset{\text{iid}}{\sim} \mathcal{D}$, then it is automatically row- and column-exchangeable, and the entries are uncorrelated. Therefore, as long as the distribution of the weights $\mathcal{D}$ satisfies the moment conditions of Definition 2, the network is in the PSEUDO-IID regime. A sufficient condition to satisfy (iii) is for $A_{ij}$ to be *sub-Gaussian* with parameter $O(n^{-1/2})$. Then, given the independence of entries, the random variable $\big|\sum_{j=1}^{n} a_j A_{ij}\big|$ would be sub-Gaussian with parameter $\|\mathbf{a}\|_2 n^{-1/2}$, and its $p^{\text{th}}$ moment is known to be $O(\|\mathbf{a}\|_2^p n^{-p/2})$; see Vershynin (2018). Appropriately scaled Gaussian and uniform IID weights, for example, meet the sub-Gaussianity criterion and therefore fall within the PSEUDO-IID class. Condition (iv) is trivial in this case.

**Orthogonal weights.** Let $A = (A_{ij}) \in \mathbb{R}^{n \times n}$ be drawn from the uniform (Haar) measure on the group of orthogonal matrices $\mathbb{O}(n)$. While the entries are not independent, they are uncorrelated, and the rows and columns are exchangeable. To verify condition (iii), we may employ the concentration of the Lipschitz functions on the sphere, since one individual row of $A$, let us say the $i^{\text{th}}$ one $(A_{i,1}, \cdots, A_{i,n})$, is drawn uniformly from $\mathbb{S}^{n-1}$. Let $f(A_{i,1}, \cdots, A_{i,n}) := \big|\sum_{j=1}^{n} a_j A_{ij}\big|$, then $f$ is Lipschitz with constant $\|\mathbf{a}\|_2$. By Theorem 5.1.4 of Vershynin (2018), the random variable $f(A_{i,1}, \cdots, A_{i,n}) = \big|\sum_{j=1}^{n} a_j A_{ij}\big|$ is sub-Gaussian with parameter $\|f\|_{\text{Lip}} n^{-1/2} = \|\mathbf{a}\|_2 n^{-1/2}$, which also implies $\mathbb{E}\big|\sum_{j=1}^{n} a_j A_{ij}\big|^p = O(\|\mathbf{a}\|_2^p n^{-p/2})$. Finally, the exact expressions of the $p^{\text{th}}$ moments are known for orthogonal matrices (see Collins et al. (2021) for an introduction to the calculus of Weingarten functions) and satisfy condition (iv). Note how we simplify the analysis here by considering square weight matrices from the second layer and onward.

## E    FURTHER NUMERICAL SIMULATIONS VALIDATING THEOREM 1

Fig. 4 shows $Q - Q$ plots for the histograms in Fig. 2 as compared to their infinite with Gaussian limit. These $Q - Q$ plots show how the PSEUDO-IID networks approach the Gaussian Process at somewhat different rates, with IID uniform approaching the fastest.

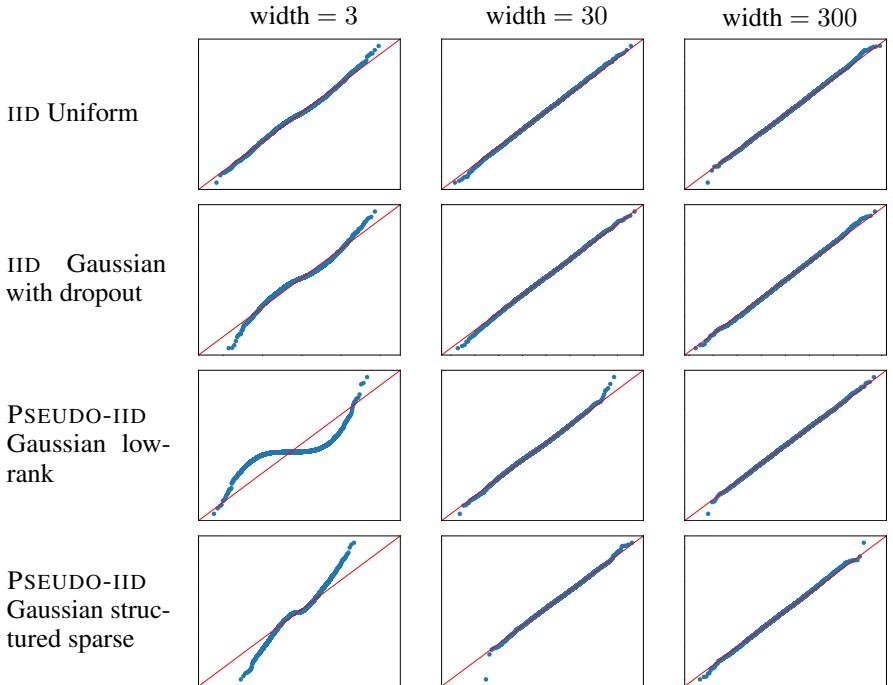

Figure 4: Q-Q plots of the preactivations values in Fig. 2 as an alternative way of showing the convergence of the preactivation of a fully connected network to a Gaussian as fully characterized in Theorem 1. The settings of the experiment are the same as those in Fig. 2.

Fig. 5 explores the growing independence of entries in $h_i^{(\ell)}(x)[n]$ for different $i$ by showing their joint distributions for two distinct choices of $i$; moreover, convergence to the limiting isotropic Gaussian distribution $(h_i^{(\ell)}(x), h_j^{(\ell)}(x))[*])$, $i \neq j$, is overlayed in the same plots. Uniform IID converges the quickest, while PSEUDO-IID Gaussian low-rank and structured sparse converge towards an isotropic distribution somewhat slower, albeit already showing good agreement at $n = 30$. The horizontal and vertical axes in each subplot of Fig. 5 are $h_i^{(5)}(x)$ for $i = 1$ and 2 respectively.

## F    NUMERICAL SIMULATIONS VALIDATING THEOREM 2 FOR ORTHOGONAL CNN FILTERS

Fig. 6 explores the growing independence of entries in $h_{i,\boldsymbol{\mu}}^{(\ell)}(\boldsymbol{X})[n]$ for different $i$ by showing their joint distributions for two distinct choices of $i$; moreover, convergence to the limiting isotropic Gaussian distribution $(h_{i,\boldsymbol{\mu}}^{(\ell)}(\boldsymbol{X})[*], h_{j,\boldsymbol{\mu}}^{(\ell)}(\boldsymbol{X})[*])$, $i \neq j$, is overlayed in the same plots.

Fig. 7 explores the asymptotic gaussian nature of entries in $h_{i,\boldsymbol{\mu}}^{(\ell)}(\boldsymbol{X})[n]$ at the same channel index $i$ by showing their joint distributions for two distinct input data. For the same reasons advocated by Garriga-Alonso et al. (2019); Novak et al. (2020), the computation of the covariance kernel comes at a high computational cost. To overcome this, we decided to focus on showing the gaussian nature of the preactivations in CNNs initialized with orthogonal filters as we defined them in 3.1. To do so, we used the Expectation-Maximization algorithm to display the level curves of the best gaussian fit to the point clouds. This gaussian turns out to have non zero correlation, in agreement with 2. Moreover, convergence to the limiting Gaussian distribution $(h_{i,\boldsymbol{\mu}}^{(\ell)}(\boldsymbol{X})[*], h_{j,\boldsymbol{\mu}}^{(\ell)}(\boldsymbol{X})[*])$, $i \neq j$, seems to be slower than in the fully connected case, if comparing this figure with Fig. 3.

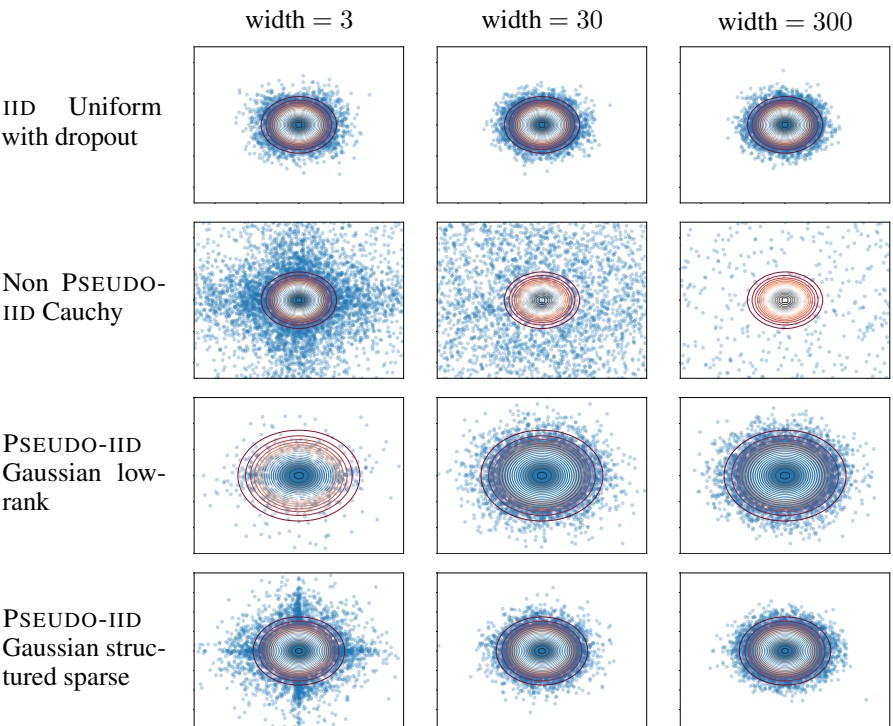

Figure 5: For fully connected networks in the PSEUDO-IID regime, it is shown in Theorem 1 that in the large width limit, at any layer, two neurons fed with the same input data become independent. We compare the joint distribution of the preactivations given in the first and second neurons at the fifth layer with an isotropic Gaussian probability density function. Initializing the weight matrices with IID Cauchy realisations falls outside of our defined framework, resulting in a poor match. The inputs were sampled from $\mathbb{S}^8$ and 10000 experiments conducted on a 7-layer network.

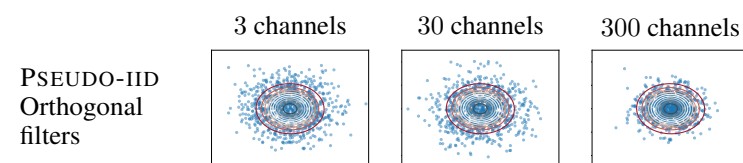

Figure 6: The empirical joint distribution of the preactivations generated by one same input flowing through a 7-layer CNN with $3 \times 3$ kernels, at two different channel indexes. An isotropic 2D gaussian is included as level curves, corroborating with the predicted asymptotic independence between neurons as given in Theorem 2. The input data $\boldsymbol{X}_a$ are drawn from the MNIST dataset and 1000 experiments were conducted. The horizontal and vertical axes in each subplot are respectively $h_{1,\boldsymbol{\mu}}^{(4)}(\boldsymbol{X}_a)$ and $h_{2,\boldsymbol{\mu}}^{(4)}(\boldsymbol{X}_a)$, where $\boldsymbol{\mu}$ is the pixel located at (5,5).

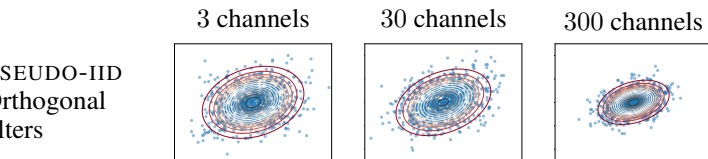

Figure 7: The empirical joint distribution of the preactivations generated by two distinct inputs flowing through a 7-layer CNN with $3 \times 3$ kernels. The large channel limiting distribution as defined in Theorem 2 involves the computation of a kernel covariance at a high cost. We use instead the Expectation-Maximization algorithm for each subplot and include as level curves the best gaussian fit. The input data $\boldsymbol{X}_a, \boldsymbol{X}_b$ are drawn from the MNIST dataset and 1000 experiments were conducted. The (correlated) gaussian nature of the preactivations is thus empirically verified as the gaussian fit goes better and better. The horizontal and vertical axes in each subplot are respectively $h_{1,\boldsymbol{\mu}}^{(4)}(\boldsymbol{X}_a)$ and $h_{1,\boldsymbol{\mu}}^{(4)}(\boldsymbol{X}_b)$, where $\boldsymbol{\mu}$ is the pixel located at (5,5).

## G  EXAMPLES CONCERNING THE BOUNDED MOMENT CONDITION (III) OF THE PSEUDO-IID DISTRIBUTION

The bounded moment condition (iii) of the PSEUDO-IID distribution in Definition 2 is a key condition in the distinct proof of IID matrices taken in Hanin (2021) (Lemma 2.9). We show some examples of distributions in Figure 8 which verify or violate the conditions we identified as sufficient to rigorously prove the convergence of random neural networks to Gaussian Processes in the large width limit.

## H  STRUCTURED SPARSE WEIGHT MATRICES IN THE FULLY CONNECTED SETTING

Fig. 9 shows examples of permuted block-sparse weight matrices used to initialize a fully connected network in order to produce the plots given in Fig. 2–5.

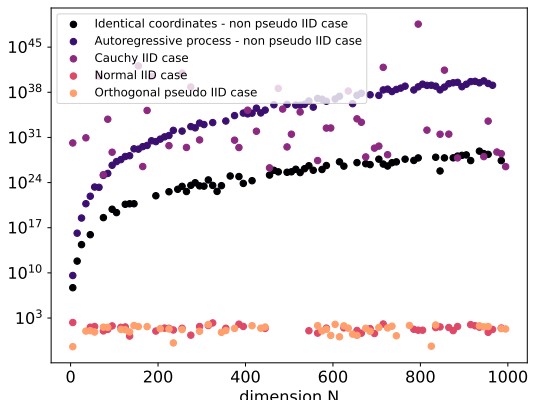

Figure 8: Plots of $\mathbb{E}\left|\sum_{j=1}^{N} X_j\right|^8$ against dimension $N$ for different cases where conditions (ii) and (iii) of the pseudo IID regime are either satisfied or violated. Condition (iii) is considered with $\mathbf{a} = (1, \cdots, 1)$ such that it becomes $\mathbb{E}\left|\sum_{j=1}^{N} X_j\right|^8 = K$, where $X = (X_1, \cdots, X_N)$ is regarded as one row of the weight matrix. The chosen distribution for the vector $X$ impacts whether the network is in the PSEUDO-IID regime. In the identical coordinates case, $X = (X_1, \cdots, X_1)$ is the concatenation of the same realisation $X_1$ sampled from a standard normal. Not only the traditional IID assumption is broken as the coordinates are obviously dependent but also condition (iii) is violated, thus resulting in an unbounded expectation when growing the dimension $N$. The autoregressive Process $(X_1, \cdots, X_N)$ shown is obtained by $X_i = \epsilon_i + X_{i-1}$, where $\epsilon_i$ are IID multivariate Gaussian noises of dimension $N$. The correlation between the coordinates does not decrease fast enough with the dimension to get a bound on the computed expectation and condition (iii) is once again violated. On the contrary, from the plot produced by sampling IID Cauchy distributions, it is not obvious whether condition (iii) holds. Nonetheless, condition (ii) which ensures the finiteness of the variance is not, thus a random network initialized with IID Cauchy weights falls outside the scope of our identified broad class of distributions to ensure a convergence towards a Gaussian Process. The last cases of samples taken either from scaled multivariate normals in red or uniformly sampled from the unit sphere in orange (with appropriate scaling such that condition (ii) holds) show that there exists a bound independent from the dimension on the expectation of interest. The empirical expectations are taken considering averages over 100000 samples.

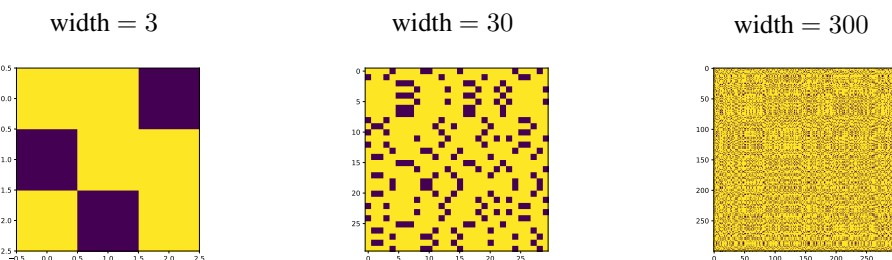

Figure 9: Example of a permuted block-sparse weight matrix at initialization of a fully connected network with increasing width $N$. The matrix is initialized with identically and independently sampled diagonal blocks from a scaled Gaussian. Its rows and columns are then randomly permuted in order to satisfy the PSEUDO-IID conditions. The block size is set to be $\lceil 0.2 * N \rceil$. Entries in yellow are zero and entries in black are nonzero.

