# OpenReview forum: "Beyond IID weights: sparse and low-rank deep Neural Networks are also Gaussian Processes"
_ICLR.cc/2024/Conference — ICLR 2024 poster_

### Official Review · Reviewer_rGFF · 2023-10-30

**Soundness:** 3 good
**Presentation:** 1 poor
**Contribution:** 2 fair
**Rating:** 5
**Confidence:** 3

**Summary:**

This paper extends the results of Matthews et al. (2018) about the convergence of random deep networks to Gaussian processes, to a wider class of distribution called ``Pseudo-IID".

**Strengths:**

The strength of this paper is yet another result/evidance about the convergence of random deep networks to a Gaussian process, for broader class of initial weight distributions.

**Weaknesses:**

The main weakness of this paper is that it seems as though that while the class of initial weight distributions is indeed broader than the Gaussian one Matthews et al. (2018), the techniques used to prove the result are pretty much the same as in Matthews et al. (2018), and the differences are just technicalities (e.g., CLT for exchangeable RVs). Therefore, since the main contribution of this paper is supposed to be for the theoretical study of NN, I believe that the paper is not novel in that sense. Also, there are many grammatical mistakes and in general it feels that the authors did not make an all out effort when writing their paper.

**Questions:**

Above I discussed my main concern with this paper. Here are some other comments:

1. Definition 2 should come before Definition 1.

2. Each of the elements in Definition 1 must be discussed right away after the definition is given. To the reader, it is almost impossible to understand why these assumptions are either needed or interesting. If it is ``difficult" to explain these at this point in the flow of the paper, then probably those definitions should come after, and keep the introduction clean from technical mathematical definitions.

3. While I read and understood the proof, it might be a good idea if the authors will explain the novelty and mathematical difficulties as compared to current literature.

---

> ### Author Response · Authors · 2023-11-17
>
> We thank the reviewer for their careful comments.
>
> **Comment:**
>
> The strength of this paper is yet another result/evidance about the convergence of random deep networks to a Gaussian process, for broader class of initial weight distributions.
>
>
> **Response:**
>
> We appreciate your skepticism about the value of another manuscript that proves the Gaussian Process convergence for a wider class of matrices without there being a clear reason for the broader class of matrices. Let us take this opportunity to highlight that the reason we derived these results was precisely because of the increasing importance of having computationally efficient networks where arguably the two most natural candidates to study are structured sparse and low-rank weight matrices. We have made a number of changes to the manuscript in order to better highlight the reason we are excited about Theorems 1 and 2.  These changes include: adding to the introduction a discussion on the computational efficiency of low-rank and structured sparse networks and the lottery ticket hypothesis recently proposed for the latter. We have also enhanced the discussion on direct practical purposes served by this Gaussian Process limit in our setting (see section 3.3). For instance, we mention the Edge of Chaos initialization strategy and how our result makes the analysis of optimal initialization of low-rank and structured-sparse networks possible in these settings for the first time. Note that orthogonally initialized CNNs have been under extensive study in the past years for their empirical enhanced trainability. To the best of our knowledge, the Gaussian Process limit has never been rigorously derived before for this case. Finally, we connected this Gaussian Process limit to bayesian neural networks (see section 3.3), underlining that we now have the tools to make *exact* Bayesian inference when initializing a network in an efficient way (sparsily or with low-rank factors), which was previously unknown.
>
> ----
>
>
>
> **Comment:**
>
> Definition 2 should come before Definition 1.
> Each of the elements in Definition 1 must be discussed right away after the definition is given. To the reader, it is almost impossible to understand why these assumptions are either needed or interesting. If it is "difficult" to explain these at this point in the flow of the paper, then probably those definitions should come after, and keep the introduction clean from technical mathematical definitions.
>
> **Response:**
>
> We have made a substantial number of changes to the manuscript based on these suggestions. We have moved the definition of exchangeability (previously Def. 2) to precede the definition of pseudo-iid (previously Def. 1). The definitions of exchangeability and pseudo-iid have also been moved to Section 2 so that the introduction has fewer technical details, and instead has a greater discussion on why we are considering these matrix ensembles. Lastly, we have included a paragraph just following Definition 2 of pseudo-iid where we discuss the reason for (i)-(iv) one by one. Thank you for these suggestions, they have improved the exposition of the manuscript.
>
> ----
>
>
>
> **Comment:**
>
> The main weakness of this paper is that it seems as though that while the class of initial weight distributions is indeed broader than the Gaussian one Matthews et al. (2018), the techniques used to prove the result are pretty much the same as in Matthews et al. (2018), and the differences are just technicalities (e.g., CLT for exchangeable RVs). Therefore, since the main contribution of this paper is supposed to be for the theoretical study of NN, I believe that the paper is not novel in that sense.
>
> **Response:**
>
> The reviewer is correct in that the proof is a natural extension of the proof strategy developed by Matthews et al. (2018), whose blueprint has widely been followed by several subsequent works, e.g. Garriga-Alonso et al. (2018) and Novak et al. (2020), to include CNNs, Huang et al. 2020 for orthogonal weights. However, as we mentioned above, the main contribution of the manuscript was not intended to be the novelty in its proof, though of course there is some novelty here, but the motivation of the manuscript is to set the foundation of rigorous theory for the more efficient and parsimonious networks using low-rank or structured sparse weight matrices.  We think this is an important contribution due to the growing widespread usage of deep networks where computational efficiency has real-world implications (see the last paragraph of the introduction). We hope the adjusted structure and re-emphasis on the motivation for the pseudo-iid class will encourage you to improve your score for the manuscript.
>
> ----

---

### Official Review · Reviewer_EeDC · 2023-11-01

**Soundness:** 3 good
**Presentation:** 4 excellent
**Contribution:** 3 good
**Rating:** 8
**Confidence:** 2

**Summary:**

The authors show that when the weights of neural networks are initialized in a "pseudo" i.i.d. manner, as defined by them, the random fields generated by each layer converge to Gaussian processes as the width of all layers goes to infinity simultaneously. They show this for fully connected neural networks and CNNs.

Their definition of pseudo i.i.d. involves row and column exchangeability, a variance given as a parameter, and bounded higher order moments. They illustrate the generality of their proposed assumption by showing several non IID examples used in practice that fall under their proposed definition.

Finally, they present numerical simulations for the convergence to Gaussian processes for fully connected networks with widths up to 300, where they examine the variance and joint distributions over a single neuron.

**Strengths:**

* Their major contribution is showing convergence of pseudo i.i.d. initializations and simultaneous scaling. This appears to be a challenging problem compared to sequential scaling, and their assumption seems quite general because it includes some non i.i.d. ways of initialization, including orthogonal and low rank weights, some of which are also faster in practice.
* Their contribution also seems like a first step to identify conditions for edge of chaos.

In general this paper is written very clearly, and the authors have given a good summary that shows exactly where their contribution fits in the existing literature. Moreover, their ideas and the proof technique for fully connected layers is easily understandable.

**Weaknesses:**

For CNNs, only one existing definition related to orthogonality, by Wang et al. in 2020, fits their pseudo i.i.d regime. Moreover I am not sure about feasibility, but including a numerical simulation for CNNs would make the study complete.

**Questions:**

Maybe it is referenced in the citations, but how will Gaussian processes in the limit help in developing new network regularisers?

---

> ### Author Response · Authors · 2023-11-17
>
> We thank the reviewer for their helpful comments and support of the manuscript.
>
> **Comment:**
>
> For CNNs, only one existing definition related to orthogonality, by Wang et al. in 2020, fits their pseudo i.i.d regime. Moreover I am not sure about feasibility, but including a numerical simulation for CNNs would make the study complete.
>
> **Response:**
>
> Thank you for suggesting considering other variants of orthogonal ensembles in CNNs. Indeed we have chosen one definition over others, as there is no consensus in the literature on how to initialize CNNs with orthogonal weights. However, the choice was not solely made based on its agreement with our pseudo-iid setting, but also because sampling from our favored ensemble can be done quite efficiently. Note that the PyTorch implementation of the orthogonal initialization of CNNs filters follows precisely this definition. Whether or not some of the other variants fit within the pseudo-iid framework remains an open question. Thanks to your remark, we have now included experiments for orthogonal CNNs (see Appendix F) that confirm our theoretical result.
>
> ----
>
>
> **Comment:**
>
> Maybe it is referenced in the citations, but how will Gaussian processes in the limit help in developing new network regularisers?
>
> **Response:**
>
> The motivation for this manuscript is to develop initialization theory for practitioners who are exploring weight matrix structures which lead to lower computational cost and/or improved accuracy by reducing the degree of overparameterization. This motivation was not properly highlighted, relegated primarily to Section 3.3 and with a brief sentence in the second paragraph on page 1. We have expanded the discussion in the second paragraph of page 1 to better explain this motivation and to cite a few of the many manuscripts exploring new matrix structures. Importantly, these new structures either satisfy the row and column exchangeability condition in Definition 2, or they are amenable to random permutations of the rows and columns due to the arbitrary column and row ordering. We have also extended the discussion of new matrix models in deep networks in Section 3.1 and have greatly increased the citations to this community.
> The regularization property that we were referring to is the observed improved test accuracy for moderately pruned or low-rank matrices such as the exciting new work on the lottery ticket hypothesis for structured sparse matrices in Chen et al. (2022) and the novel approach in Breyland et al. (2023), where during training the matrices have their rank reduced by singular value thresholding based on the upper limit of the Marchenko-Pastur distribution which is suggestive of unlearned components which again shows an improvement in accuracy due to the reduced overparameterization.
>
>
> ----

---

### Official Review · Reviewer_SWNp · 2023-11-01

**Soundness:** 2 fair
**Presentation:** 3 good
**Contribution:** 2 fair
**Rating:** 6
**Confidence:** 3

**Summary:**

Generalises previous results regarding neural networks with IID initialization becoming Gaussian Processes (GP) in the infinite-width limit to other initialization schemes, such as orthogonal, normalised, low-rank and sparse initialization schemes. Prove the results for both fully connected and CNN networks and provide simulation results for the different initialization schemes.

**Strengths:**

* The presented PSEUDO-IID framework replaces the IID requirement with centered and uncorrelated  entries, in a row-exchangable and column-exchangable matrix, under some technical conditions to prevent dependencies from changing the results. This is a general alternative to IID, which may be a useful generalisation for other cases as well.
 * Many different initialization schemes are PSEUDO-IID: IID, orthogonal, low-rank, and permuted block-sparse.

**Weaknesses:**

* The weights of the first layer must be Gaussian IID in all the relevant cases.

**Questions:**

* Is it correct that the weights of the first layer must be Gaussian IID in all the relevant cases? So the proof work only for PSEUDO-IID weights in all other layers?

---

> ### Author Response · Authors · 2023-11-17
>
> We thank the reviewer for their helpful comments.
>
> **Comment:**
>
> Is it correct that the weights of the first layer must be Gaussian IID in all the relevant cases? So the proof work only for PSEUDO-IID weights in all other layers?
>
> **Response:**
>
>
> Thank you for raising this interesting point.  In our proof of convergence at layer $\ell+1$, the argument relies on the independence of the prior layer's output *in the limit*. Since the input dimension $N_0$ is fixed and not scaled with $n$, the distribution of the first layer's output remains unchanged with scaling the network, therefore we need to make sure $(z^{(1)}_{i})$ are independent by forcing the rows of $W^{(1)}$ to be iid. If $N_0$ were to go to infinity, we wouldn't need such consideration.
>
>
> We state Theorems 1 and 2 with
> $W^{(1)}$ having Gaussian iid entries for simplicity so that the first layer would also be a Gaussian Process, however, iid rows in the first layer are sufficient to prove GP limit for layer two onward. Examples of row iid distributions that could be used for $W^{(1)}$ which are similar to the low-rank and structured ensembles include the product $W^{(1)}=DPC$ of a diagonal matrix $D\in \mathbb{R}^{N_{1}[n]\times N_{1}[n]}$ with entries drawn iid, $P$ a random permutation matrix, and $C\in \mathbb{R}^{N_{1}[n] \times N_{0}}$ a non-zero low-rank or structured sparse matrix. This is now stated on page 4 as a footnote and also in the appendix.
>
> We emphasize that although the extra assumption about the first layer is essential to our proof, our experiments are conducted with pseudo-iid weights at all layers (including the first one), and they show the very convergence to a Gaussian Process limit that we expect. We conjecture that the effect of the first layer will become negligible in deeper networks, as typically the covariance kernels converge (in depth) to some limiting fixed point.
>
> We have made this point clearer in our revised manuscript, which also benefited from the other reviewers' feedback. We hope the changes we have made will encourage you to improve your score for the manuscript.

---

> > ### Comment · Reviewer_SWNp · 2023-11-20
> > **Response to authors**
> >
> > I'd like to thank the authors for their detailed responses to the reviewers' comments. I believe there is a consensus between the authors and the reviewers about the contribution of the paper. Accepting the paper or not boils down to the question posed by reviewer @rGFF, asking if the novel "technicalities" used in extending a previous proof (Matthews et al. (2018)) to a larger class of initialization choices are enough in terms contribution and future impact. @EeDC says yes, @rGFF says no and I'm not sure.
> >
> > The main reason for acceptance (for me) is the hope that those "technicalities" would make impact on *other*, unrelated, problems where IID assumption may be relaxed. With this hope I will increase my score to marginally above acceptance rate and would leave it for the AC to make the final call.

---

> > > ### Author Response · Authors · 2023-11-21
> > >
> > > Thank you for your answer. Indeed relaxing the independence assumption while allowing some vanishing dependencies is a general approach with potential impact in other contexts. Thank you for pointing this out.

---

### Official Review · Reviewer_uyEL · 2023-11-02

**Soundness:** 3 good
**Presentation:** 3 good
**Contribution:** 3 good
**Rating:** 6
**Confidence:** 3

**Summary:**

The paper extends previous results that showed randomly initialized multi-layer Bayesian neural networks with i.i.d. parameters are equivalent to Gaussian processes at the infinite width limit. The paper improves upon existing results by showing a similar result for a broader class of random parameters, namely pseudo-iid parameters. This class of random variables not only subsume classes of random variables that were included in previous literature (iid, orthogonal), but include new ones such as low-rank and block-sparse random variables. The authors results apply to fully connect networks as well as convolutional neural networks.

**Strengths:**

- The connections to previous literature and the contributions of the present work in light thereof are clearly stated.
- The writing and exposition is clear and easy-to-follow.
- The new class of random variables for which the authors extend existing results is significantly larger than those investigated previously, and can facilitate future research in multiple directions.

**Weaknesses:**

- The paper expands upon existing results in a fairly specific strand of research, and it is not immediately clear why the novel families of random variables that they confirm to constitute Gaussian processes would be practically interesting. Though they provide some justifications in Section 3.3, these ideas are left for future research.
- Their focus on randomly initialized deep neural networks in general, rather than Bayesian neural networks in Matthews et al. 2018 also limits the practical implications of their work. Though the latter were able to examine the implications of BNN-GP's in posterior inference, this is out-of-scope for the current work, limiting its practical implications to initialization schemes.

**Questions:**

- Pg. 1, Definition 1: $\mathbf{a}$ is not defined.
- Pg. 2, Section 1.1: It might make sense to mention the focus of some previous work on Bayesian neural networks, and how/why this is not the case in the current paper.
- Pg. 3, Section 2.1: Referring to $N_0$ as the width of the first layer might be confusing, given that the input layer is different than the layer $l=1$, which might also be considered the first layer.
- Pg. 3: What is the significance of the input space being assumed countably infinite?
- Pg. 6, Section 3.1: It would make sense to be more explicit about how each of the following example distributions are not covered by previous research's findings.

===========

Post-rebuttal note: I thank the authors for their feedback. Although the arguments for the paper's contributions being limited have some merit, I still believe the contributions are sufficient to warrant acceptance, and retain my recommendation for doing so.

---

> ### Author Response · Authors · 2023-11-17
>
> We thank the reviewer for their careful comments. We incorporated your feedback regarding the presentation and wording. Regarding your other questions:
>
> **Comment:**
>
> Pg. 1, Definition 1: a is not defined.
>
> **Response**
>
> Thank you for raising this omission.  Definitions 2 (previously 1) and 4 have been edited to state that this is true for any fixed vector a of the appropriate size.
>
> ----
>
>
> **Comment:**
>
> [I]t is not immediately clear why the novel families of random variables that they confirm to constitute Gaussian processes would be practically interesting. Though they provide some justifications in Section 3.3, these ideas are left for future research.
>
> **Response**
>
> We have expanded the discussion in the introduction on the practical significance of deep neural networks with low-rank and sparse structured matrices, which are special instances of our pseudo-iid regime. These instances are good candidate for reducing the number of parameters of very large models while retaining good accuracy. Recently, the lottery ticket hypothesis has empirically been shown for structured sparse networks and we anticipate such findings could be made in the low rank case as well. We give in the manuscript numerous reasons why practitioners would want to use such networks instead of their dense/full-rank counterparts. This being said, studying these winning tickets at initialization is a first step towards a better understanding of their training dynamics as the structured sparse and pruning communities lack rigorous initialization theory, upon which we expanded in Section 3.3 (paragraph Edge of Chaos).
>
> ----
>
> **Comment:**
>
> Section 1.1: It might make sense to mention the focus of some previous work on Bayesian neural networks, and how/why this is not the case in the current paper.
>
> **Response:**
>
> Thank you for raising this omission.  We had not considered this use case of the results.  Our results are also applicable to BNN-GP which we have now included in a paragraph of Section 3.3. Exactly as in the iid case, the Gaussian Process limit in pseudo-iid regimes gives an exact prior based on which exact Bayesian inference can be made. We connected our work to the Bayesian framework, as suggested.
>
> ----
>
>
> **Comment:**
>
> What is the significance of the input space being assumed countably infinite?
>
>
> **Response:**
>
> We show in this paper a convergence in distribution with respect to a certain topology, induced by the metric given in Equation (11). It is not clear how to define at first this convergence of stochastic processes indexed by an uncountable set. Alternatively, following another proof strategy to show the Gaussian Process limit for iid weights, the input space is assumed to be compact in Hanin (2021). In fact, we regard the data as a fixed point cloud, as opposed to coming from a certain distribution.
>
> ----
>
>
>
> **Comment:**
>
> It would make sense to be more explicit about how each of the following example distributions are not covered by previous research's findings.
>
> **Response:**
>
> At the start of Section 3.1 we have clarified which results are novel and which have previously been proven.  In short, the Gaussian Process limit has been proven for iid and orthogonal matrices for fully-connected networks and only iid convolutional networks.To further emphasize the novel distributions we have moved the case of iid and orthogonal fully-connected to Appendix D, but have retained references to these prior contributions at the start of Section 3.1.  We hope this change helps highlight the new examples of pseudo-iid which motivated this manuscript.
>
> ----
>
> Altogether, to address your concerns, we have added a section on BNN-GP, emphasized the practical significance of the pseudo-iid regime and the implications of the GP limit in these cases and clarified the novelty of our work. We hope the reviewer is content with this improved version of the manuscript and will consider improving their score.

---

### Author Response · Authors · 2023-11-21
**Follow-up on review responses**

Thank you to all reviewers once again for your valuable time and insightful suggestions so far which have helped us improve the paper further.  We hope you will be able to look at our changes and answers (although we know there is not much time left for you), from which you could consider adjusting our scores accordingly. We will be happy to reply promptly to any further comments you have.

---

### Meta-Review · Area_Chair_PBZt · 2023-12-11

**Metareview:**

The paper presents an extension of the Gaussian Process limit to a broader class of initial weight distributions in deep neural networks, notably including sparse and low-rank structures.

The reviewers, overall, acknowledge the paper's technical soundness and clarity in presentation. The extension to pseudo-i.i.d. parameters and the application to both fully connected and convolutional neural networks are significant strengths.

There were concerns including regarding the practical relevance of the novel families of random variables. The authors addressed most of the reviewers' concerns.

**Justification For Why Not Higher Score:**

Given the reviews, a spotlight is not adequate.

**Justification For Why Not Lower Score:**

The paper adds to the theoretical framework of neural network initialization and opens avenues for further research,

---

### Decision · Program_Chairs · 2024-01-16

Accept (poster)